# Non-Asymptotic and Non-Lipschitzian Bounds on Optimal Values in Stochastic Optimization Under Heavy Tails

**Jindong Tong** [1]  **Hongcheng Liu** [2]  **Johannes O. Royset** [3]

## Abstract

This paper focuses on non-asymptotic confidence bounds (CB) for the optimal values of stochastic optimization (SO) problems. Existing approaches often rely on two conditions that may be restrictive: The need for a global Lipschitz constant and the assumption of light-tailed distributions. Beyond either of the conditions, it remains largely unknown whether computable CBs can be constructed. In view of this literature gap, we provide three key findings below: (i) Based on the conventional formulation of sample average approximation (SAA), we derive non-Lipschitzian CBs for convex SO problems under heavy tails. (ii) We explore diametrical risk minimization (DRM)—a recently introduced modification to SAA—and attain non-Lipschitzian CBs for nonconvex SO problems in light-tailed settings. (iii) We extend our analysis of DRM to handle heavy-tailed randomness by utilizing properties in formulations for training over-parameterized classification models.

## 1. Introduction

Consider a stochastic optimization (SO) problem formulated as below:

$$\underset{\mathbf{x} \in \mathcal{X}}{\text{minimize}}\ F(\mathbf{x}) := \mathbb{E}[f(\mathbf{x}, \xi)], \tag{1}$$

where $\mathcal{X} \subseteq \mathbb{R}^d$ is a nonempty, bounded, and closed feasible region, $\xi$ is a random vector of problem parameters

[1]Department of Industrial and Systems Engineering, University of Florida, Gainesville, 32611 USA. Email: jindong-tong@ufl.edu [2]Department of Industrial and Systems Engineering, University of Florida, Gainesville, 32611 USA. Email: liu.h@ufl.edu [3]Department of Industrial and Systems Engineering, University of Southern California, Los Angeles, CA 90089. Email: royset@usc.edu. Correspondence to: Hongcheng Liu <liu.h@ufl.edu>.

*Proceedings of the $42^{nd}$ International Conference on Machine Learning*, Vancouver, Canada. PMLR 267, 2025. Copyright 2025 by the author(s).

with probability distribution $\mathbb{P}$ supported on $\Xi \subseteq \mathbb{R}^m$. The function $f : \mathbb{R}^d \times \Xi \to \mathbb{R}$ is assumed to be deterministic and measurable, where $d$ denotes problem dimensionality. We further assume the expectation $\mathbb{E}[f(\mathbf{x}, \xi)] = \int_{\Xi} f(\mathbf{x}, \xi)\, d\mathbb{P}(\xi)$ to be well-defined and finite-valued for every $\mathbf{x} \in \mathcal{X}$. The SO problems of the above have been much discussed in literature (e.g. by Shapiro et al., 2021; Ruszczyński & Shapiro, 2003; Lan, 2020; Royset & Wets, 2021) and has wide applicability including in machine learning (as per, e.g., Bartlett et al., 2006). Throughout this paper, we let $F^*$ be the infimum of $F(\mathbf{x})$ over $\mathcal{X}$.

Regarding such SO problems, the focus of this paper is to construct computable non-asymptotic confidence bounds (NCB) for the value of $F^*$ when given an i.i.d. sample $\xi_1, ..., \xi_N$ of $\xi$; namely, for a large sample size $N$ and a significance level $\alpha \in (0, 1)$, we are to find the values of the two scalars, denoted by $L_{N,\alpha}$ and $U_{N,\alpha}$, such that the following relationship holds:

$$\text{Prob}\left[L_{N,\alpha} \leq F^* \leq U_{N,\alpha}\right] \geq 1 - \alpha. \tag{2}$$

Tight estimates of these two scalars are of usefulness. For instance, according to Guigues et al. (2017), these bounds can be utilized to qualify the accuracy of approximate solutions so as to construct stopping criteria for stochastic algorithms. Other potential applications include statistical decisions in computing confidence intervals and testing statistical hypotheses about the optimal value as well as in designing algorithms for multi-armed bandit problems.

Following many discussions on confidence bounds (as per, e.g., Kaniovski et al., 1995; King & Rockafellar, 1993; Pflug, 2003; Shai et al., 2009; Shapiro, 2003; Shapiro & Nemirovski, 2005), this paper is focused on their construction through sample average approximation (SAA) summarized as below. Let $F_N(\mathbf{x}) := N^{-1} \sum_{j=1}^{N} f(\mathbf{x}, \xi_j)$ and define

$$\underset{\mathbf{x} \in \mathcal{X}}{\text{minimize}}\ F_N(\mathbf{x}), \tag{3}$$

and denote its optimal cost by $F_N^*$. Then, many existing NCBs in (2) are constructed in the form of

$$L_{N,\alpha} := F_N^* - l_{N,\alpha};$$
$$U_{N,\alpha} := F_N^* + u_{N,\alpha},$$

for some quantities $l_{N,\alpha}$ and $u_{N,\alpha}$ that are usually vanishing in $N$, for all $\alpha$. Some of our results (such as for nonconvex SO problems) also consider a recent variation to (3) called the diametrical empirical risk minimization (DRM) introduced by Norton & Royset (2023) and formulated as

$$F_\gamma^* := \inf_{\mathbf{x} \in \mathcal{X}} \sup_{\mathbf{v}:\|\mathbf{v}\| \leq \gamma} F_N(\mathbf{x} + \mathbf{v}), \qquad (4)$$

where $\gamma \geq 0$ is the diametrical risk radius, and $\|\cdot\|$ is any norm used to define the neighborhood.

Among the many results on confidence bounds, earlier discussions focus mostly on asymptotic settings (as per, e.g., Dupacová & Wets, 1988; King & Rockafellar, 1993; Mak et al., 1999; Pflug, 1995; 1999; 2003). In contrast, of our focus in this paper is the non-asymptotic settings comparable to the discussions by the recent works of, e.g., Guigues et al. (2017); Oliveira & Thompson (2023).

Nonetheless, to our knowledge, almost all existing non-asymptotic results are subject to at least one of the following limitations:

- Existing NCB bounds (such as by Oliveira & Thompson, 2023) may (sometimes drastically) deteriorate with the growth of the Lipschitz constant of $F$.

- Existing benchmark NCBs (as in Guigues et al., 2017), while being free from the Lipschitz constant of $F$, stipulate light-tailed assumptions; namely, the $p$th moment of the underlying randomness should exist for all $p \geq 1$. The same work by Guigues et al. (2017) further assumes convexity in the SO formulation.

Note that, when $p = 2$, counterpart NCBs have been rigorously constructed through the solutions generated by a stochastic mirror descent method (as in Lan et al., 2012; Guigues, 2017). Nonetheless, the resulting NCBs are still Lipschitz constant-dependent. Meanwhile, the evolution of the bounds with varying values of $p$ have not been explicated for any finite $p > 2$. In view of these limitations, the main question of this research is as below:

**Main Research Question:** *Can NCBs be constructed through the SAA or DRM in scenarios beyond the simultaneous assumptions of a known global Lipschitz constant and light-tailed underlying randomness?*

To this question, this paper presents perhaps the first set of affirmative answers summarized as below:

*Main result 1:* As shown in Theorem 4.1, for convex SO problems with a differentiable expected cost function $F$, we show, perhaps for the first time, that the non-Lipschitzian NCBs can be derived in heavy-tailed settings with quantities

in (2) explicated into:

$$
\begin{aligned}
L_{N,\alpha} &= F_N^* - \frac{2.74}{\alpha^{1/p}} \cdot 6^{1/p} \cdot \sigma_f \cdot \sqrt{\frac{p}{N}}; \quad \text{and} \\
U_{N,\alpha} &= F_N^* + \frac{2.74}{\alpha^{1/p}} \cdot 6^{1/p} \cdot (\sigma_g \mathcal{D}_q + \sigma_f) \cdot \sqrt{\frac{p}{N}},
\end{aligned}
\tag{5}
$$

where $p$ is the highest order of existent central moments, which are assumed equal to $\sigma_f^p$ and $\sigma_g^p$, of the underlying randomness and $\mathcal{D}_q$ is the diameter of feasible region. These bounds exhibit the same rate with $N$ as in the results of Guigues et al. (2017) and is often sharper than the bounds by Oliveira & Thompson (2023) in terms of the dependence on dimensions $d$ and the global Lipschitz constant.

*Main result 2:* Under nonconvexity and light tails, we further derive NCBs through combining SAA and DRM and provide the following specification of (2):

$$
\begin{aligned}
L_{N,\alpha} &= F_N^* - 4\sigma_\psi \sqrt{\frac{1}{2N} \cdot \ln\frac{2}{\alpha}} + \frac{4\sigma_\psi}{N} \cdot \left(\ln\frac{2}{\alpha}\right); \quad \text{and} \\
U_{N,\alpha} &= F_\gamma^* + 4\sigma_\psi \sqrt{\frac{1}{2N}} \cdot \sqrt{\frac{\Delta_{p,q} d^{2/q}}{\gamma^2} + \ln\frac{2}{\alpha}} \\
&\quad + \frac{4\sigma_\psi}{N} \cdot \left(\frac{\Delta_{p,q} d^{2/q}}{\gamma^2} + \ln\frac{2}{\alpha}\right),
\end{aligned}
\tag{6}
$$

where $p, q \geq 1$ are any admissible scalars satisfying $1/p + 1/q = 1$, $\sigma_\psi$ is a variance-like term related to the sub-exponential assumption on the underlying randomness, and $\Delta_{p,q} \approx O(\max_{\mathbf{x} \in \mathcal{X}} \|\mathbf{x}\|_p^2)$ is comparable to the $p$-norm diameter of $\mathcal{X}$. With further interpretation (to be shown in formal result), when the the scope of the problem is fixed within a bounded 1-norm unit ball, $\Delta_{p,q}$ also grows almost linearly in $q$, which will lead to a $O(\ln d)$ dependence on dimensionality in $U_{N,\alpha}$. To our knowledge, the above is the first non-Lipschitzian NCB result for nonconvex SO problems.

*Main result 3.* In Theorem 4.13, we further extend *Main result 2* to heavy-tailed settings after imposing some plausible structural assumptions that $f$ is everywhere non-negative and the DRM yields a zero optimal cost — as often is the case for over-parameterized machine learning models. We

show that

$$L_{N,\alpha} = F_N^* - \frac{2.74}{\alpha^{1/p}} \cdot 4^{1/p} \cdot \sigma_f \cdot \sqrt{\frac{p}{N}}; \quad \text{and}$$

$$U_{N,\alpha} = \left[ F_\gamma^* + \frac{8}{\ln 2} \cdot \left( \frac{\Delta_{p,q} d^{2/q}}{\gamma^2} + \ln \frac{4}{\alpha} \right) \right.$$
$$\left. \cdot \left( 1 + \frac{\sigma_f'}{p-1} \right) \cdot N^{-\frac{p-2}{2p}} \right]$$
$$\cdot \left( 1 - \frac{7}{\ln 2} \cdot \left( \frac{\Delta_{p,q} d^{2/q}}{\gamma^2} + \ln \frac{4}{\alpha} \right) \cdot N^{-\frac{1}{2}} \right)^{-1},$$

$$\tag{7}$$

where $p$ is the highest order of existent central moment for the underlying randomness. As mentioned in Main result 2, $\Delta_{p,q}$ grows almost linearly with $q$; namely $\Delta_{p,q} \leq C \cdot q \cdot \max_{\mathbf{x} \in \mathcal{X}} \|\mathbf{x}\|_p^2$ for some universal constant $C$. To our knowledge, (7) is the first non-Liphschitzian NCB result for nonconvex and heavy-tailed SO.

Collectively, this paper presents a new set of non-Lipschitzian NCBs for SO problems, providing the capability of handling heavy tails in all convex cases and important instances of nonconvex cases.

## 1.1. Organizations

The rest of this paper is organized as follows: Section 2 summarizes related works. Section 3 discusses preliminaries and assumptions. Our main theorems (Theorem 4.1 for convex SAA-based NCBs, and Theorems 4.7 and 4.13 for nonconvex DRM-based NCBs) are presented in Section 4. Finally, Section 6 concludes the paper.

## 1.2. Notations

Denote by $\mathbb{R}$ the collection of all real numbers, and by $\mathbb{R}_+$ that of the non-negative ones. For any vector $\mathbf{v} = (v_1, \ldots, v_d)^\top \in \mathbb{R}^d$, denote by $\|\mathbf{v}\|_p := (\sum_{i=1}^d |v_i|^p)^{1/p}$ the $p$-norm ($p \geq 1$, finite). We also define the its infinity norm $\|\mathbf{v}\|_\infty := \max\{|v_1|, \ldots, |v_d|\}$ Meanwhile, we define the $L^p$-norm of a random vector $\boldsymbol{\zeta} = (\zeta_i) \in \mathbb{R}^d$ to be $\|\boldsymbol{\zeta}\|_{L^p} := (\sum_{i=1}^d \mathbb{E}_{\zeta_i}[|\zeta_i|^p])^{1/p}$. We let $\partial F(\mathbf{x})$ and $\partial_{\mathbf{x}} f(\mathbf{x}, \xi)$ be the subdifferential of $F$ and $f(\cdot, \xi)$ w.r.t. $\mathbf{x}$. Here, we use $|v|$ to denote the absolute value of $v$ if it is a real number; otherwise, $|\mathcal{V}|$ is the cardinality of $\mathcal{V}$, when it is a set. Let $\Gamma(\cdot)$ be the Gamma function. The expectation operator is denoted by $\mathbb{E}[\cdot]$, with the underlying probability distribution understood from the context.

## 2. Related Work

Existing results on confidence bounds can be categorized into asymptotic and non-asymptotic ones, and most existing non-asymptotic finite sample results assume light-tailed underlying randomness. As mentioned, asymptotic results have been made available by Dupacová & Wets (1988); King & Rockafellar (1993); Mak et al. (1999); Pflug (1995; 1999; 2003); Duchi et al. (2021). However, as commented by Guigues et al. (2017), asymptotic analysis of such may be unreliable in some applications. In contrast, non-asymptotic results — as is the focus of this paper — can often be more informative, particularly for a practically affordable/accessible sample size.

However, NCB results are much less visited in literature. Among the scarcely available results, Guigues et al. (2017) and Oliveira & Thompson (2023) provide state-of-the-art benchmarks. In particular, for SAA in convex SO under sub-Gaussian assumptions, Guigues et al. (2017) explicate $L_{N,\alpha}$ and $U_{N,\alpha}$ of (2) into the below:

$$L_{N,\alpha} = F_N^* - \widetilde{\mathcal{O}} \left( \sqrt{\frac{\ln(1/\alpha)}{N}} \right); \quad \text{and}$$

$$U_{N,\alpha} = F_N^* + \widetilde{\mathcal{O}} \left( \sqrt{\frac{\ln(1/\alpha)}{N}} + \frac{\ln(1/\alpha)}{N^{3/2}} \right),$$

$$\tag{8}$$

where $\widetilde{\mathcal{O}}(\cdot)$ hides all other quantities that do not depend on "·" or problem dimensions $d$. Notably, these bounds are in absence of metric entropy terms (e.g., in the form of the logarithm of the covering number) for the feasible region. Quantities that reflect the feasible region's metric entropy are common in other related non-asymptotic analyses of SAA's performance (such as by Shapiro et al., 2021). Their presence often leads to a less desirable growth rate with $d$. Another desirable feature in the results of Guigues et al. (2017) is their independence on the Lipschitz constant, which is often hard to estimate and potentially hard to control in many applications.

Beyond light-tail assumptions, NCBs are provided by Oliveira & Thompson (2023) and Lan et al. (2012) through different computational schemes. More specifically, Oliveira & Thompson (2023) focus on NCBs based on solutions to the SAA formulation and show that, when the underlying randomness admits a finite $p$th moment, the quantities in (2) could be explicated as

$$L_{N,\alpha} = F_N^* - \widetilde{\mathcal{O}} \left( M_p \left( \sqrt{\frac{\ln(1/\alpha)}{N}} + \frac{\gamma(\mathcal{X})}{\sqrt{N}} \right) \right); \quad \text{and}$$

$$U_{N,\alpha} = F_N^* + \widetilde{\mathcal{O}} \left( M_p \left( \sqrt{\frac{\ln(1/\alpha)}{N}} + \frac{\gamma(\mathcal{X})}{\sqrt{N}} \right) \right),$$

$$\tag{9}$$

for a sufficient sample size of $N \geq \widetilde{O}\left(\alpha^{-2/p}\right)$. Herein, $\gamma(\mathcal{X})$ is the generic chaining functional, which quantifies the metric entropy of $\mathcal{X}$. Meanwhile, $(M_p)^{2p}$ is the $p$th moment for the squared Lipschitz constant of $f(\cdot, \xi)$. Be-

cause $\gamma(\mathcal{X})$ grows at $O(\sqrt{d})$ with increasing problem dimensions, the results in (9) would become less appealing in a higher dimensional region. Furthermore, unlike the results by Guigues et al. (2017), the results in (9) grow polynomially with the Lipschitz constant. Nonetheless, the above results by Oliveira & Thompson (2023) generally apply to both convex and nonconvex SO problems, while the discussions by Guigues et al. (2017) are mainly limited to scenarios under convexity.

Meanwhile, Lan et al. (2012) construct NCBs for convex SO problems through mirror descent stochastic approximation (MDSA), which leads

$$L_{N,\alpha} = LB^N - \widetilde{\mathcal{O}}\left(M_2 \cdot \sqrt{\frac{1}{N\alpha}}\right); \quad \text{and}$$

$$U_{N,\alpha} = F_N^* + \widetilde{\mathcal{O}}\left(\sqrt{\frac{1}{N\alpha}}\right),$$

where $(M_2)^2$ is the second moment of the Lipschitz constant of $f(\cdot, \xi)$ and $LB^N$ is a first-order Taylor expansion-based computable quantity constructed using historical solutions and sample parameters generated throughout the algorithm iterations. Note here that a light-tailed version of NCBs are also provided by Lan et al. (2012), yet their results are comparably less appealing than (8) due to higher dependence on the Lipschitz constant. To our knowledge, for both convex and nonconvex SO problems, there exists no NCB results that are simultaneously non-Lipschitzian and applicable to heavy-tailed assumptions. Especially for the nonconvex case, light-tailed but non-Lipschitzian NCBs are also in lacking.

Our results in the nonconvex case are based on the DRM formulation, which is a recent modification of SAA introduced by Norton & Royset (2023); Foret et al. (2021); Kwon et al. (2021); Zheng et al. (2020) under different names. In contrast to distributionally robust optimization (DRO), which selects a robust solution from all distributions within a prescribed ambiguity set, DRM aims to minimize the maximum empirical risk within a small neighborhood of decision variables. Via the minimax structure, DRM is capable of avoiding sharp minimizers that have comparatively higher sample average cost (a.k.a., empirical risk) in the neighborhood and, thus, should lead to better robustness and out-of-sample performance, as by McCollum et al. (2023). The underlying mechanism of minimizing sharpness through DRM and the various notions of sharpness in the objective landscape are discussed by Wen et al. (2023). Theoretical advantages of DRM are identified by Norton & Royset (2023) and Tsai et al. (2021). Meanwhile, experiments in using DRM to train machine learning models (as per, e.g., Norton & Royset, 2023; Zheng et al., 2020; Foret et al., 2021; Kwon et al., 2021) in computer vision as well as other applications (e.g., Wen et al., 2023), have supported DRM's strong empiri-

cal performance, particularly when noisy labels are present in the training data. To compute the DRM formulation, practically effective methods based on stochastic gradient descent (SGD) have also been derived by Norton & Royset (2023); Zheng et al. (2020); Foret et al. (2021), and Kwon et al. (2021). Among them, sharpness-aware minimization (SAM) methods are introduced by Foret et al. (2021) and Kwon et al. (2021). Accelerated implementations of SAM is studied by Du et al. (2022).

Similar but critically different methods than DRM include adversarial training (Bai et al., 2021) which perturbs the sampled parameters (namely, the values of $\xi_j$'s) and sometimes both the sampled parameters and the decision variables, as by Wu et al. (2020). The DRM also falls within the broader studies of robust decisions in optimization, for which the readers are referred to Lewis & Pang (2010); Men et al. (2014).

## 3. Assumptions and Preliminaries

Let $\delta \geq 0$, denote that $\boldsymbol{\xi}_1^N := (\xi_1, ..., \xi_N)$ to be a random vector with i.i.d. entries, and assume the existence of a measurable function $\widehat{\mathbf{x}}_\delta : \Xi^N \rightarrow \mathcal{X}$ such that $F_N\left(\widehat{\mathbf{x}}_\delta(\boldsymbol{\xi}_1^N)\right) \leq \inf_{\mathbf{x} \in \mathcal{X}} F_N(\mathbf{x}) + \delta$; namely, $\widehat{\mathbf{x}}_\delta$ is a $\delta$-suboptimal solution to the SAA formulation. (Hereafter, we often use the shorthand notation that $\widehat{\mathbf{x}}_\delta := \widehat{\mathbf{x}}_\delta(\boldsymbol{\xi}_1^N)$.) Its existence can be inferred by the measurability results of optimal solutions to SAA as per Shapiro et al., 2021; Rockafellar & Wets, 1998; Krätschmer, 2023). Similarly, $\widehat{\mathbf{x}}_{\delta,\gamma} := \widehat{\mathbf{x}}_{\delta_\gamma}(\boldsymbol{\xi}_1^N)$ is a $\delta$-optimal solution to the DRM formulation, namely, $\sup_{\mathbf{v}:\|\mathbf{v}\| \leq \gamma} F_N\left(\widehat{\mathbf{x}}_{\delta,\gamma}(\boldsymbol{\xi}_1^N) + \mathbf{v}\right) \leq \inf_{\mathbf{x} \in \mathcal{X}} \sup_{\mathbf{v}:\|\mathbf{v}\| \leq \gamma} F_N(\mathbf{x} + \mathbf{v}) + \delta$. We further denote by $\mathcal{X}^*$ the set of minimal solutions to the SO problem (1).

Our results impose different combinations of the following assumptions of underlying randomness.

**Assumption 3.1.** For a given $p \geq 2$, there exist an optimal solution $\mathbf{x}^* \in \mathcal{X}^*$ and a measurable function $g_f^* : \Xi \rightarrow \mathbb{R}^d$ such that the following holds: (i) $g_f^*(\xi) \in \partial f(\mathbf{x}^*, \xi)$ with $\mathbb{E}[g_f^*(\xi)] \in \partial F(\mathbf{x}^*)$; and (ii) it further holds that:

$$\|g_f^*(\xi) - g_F^*\|_{L^p} \leq \sigma_g, \quad \forall g_F^* \in \partial F(\mathbf{x}^*),$$

for some $\sigma_g \in \mathbb{R}_+$.

Assumption 3.1 stipulates the $p$-th central moment of $\nabla f(\cdot, \xi)$ to be bounded only at least for an optimal solution $\mathbf{x}^*$. With the potential non-existence for $(p+1)$-th central moment, results built on Assumption 3.1 (e.g., as in Theorem 4.1 below) could be interpreted as under heavy tails. This assumption is weaker than the common light-tailed assumptions imposed everywhere on $\mathcal{X}$ (such as by Guigues et al., 2017).

*Remark* 3.2. The Lipschitz constant of a (population-level formulation of) stochastic program is associated with the norm of the gradient of the expected cost function. Assumption 3.1 imposes upper bound on the central moment of the gradient of the random cost function. One can easily construct cases where the Lipschitz constant grows while $\sigma_g$ remains unchanged.

**Assumption 3.3.** For some $p \geq 2$ and $\sigma_f > 0$, it holds that

$$\|f(\mathbf{x}^*, \xi) - F(\mathbf{x}^*)\|_{L^p} \leq \sigma_f,$$

for any $\mathbf{x}^* \in \mathcal{X}^*$.

*Remark* 3.4. Our results on convex SO problems assume the combination between both Assumptions 3.1 and 3.3. This combination is our weakest assumption as per the underlying randomness. Benchmark results by Guigues et al. (2017) require both the random cost function $f(\mathbf{x}, \xi)$ and the stochastic (sub)gradient $\nabla f(\mathbf{x}, \xi)$ to be light-tailed for every $\mathbf{x} \in \mathcal{X}$. In contrast, the stipulation of the said combination serves for heavy-tailed and localized alternatives to the assumptions imposed by Guigues et al. (2017).

For our results on nonconvex SO, some of our results impose the following less flexible but still heavy-tailed assumption on the underlying randomness.

**Assumption 3.5.** For some $p \geq 2$ and $\sigma_f' > 0$, it holds that

$$\|f(\mathbf{x}, \xi) - F(\mathbf{x})\|_{L^p} \leq \sigma_f',$$

for any $\mathbf{x} \in \mathcal{X}$.

*Remark* 3.6. As a stronger version of Assumption 3.3, Assumption 3.5 stipulates the $p$-th central moment of $f(\cdot, \xi)$ is bounded for all $\mathbf{x} \in \mathcal{X}$. Despite additional stringency, this assumption permits a uniform bound over $\mathbf{x}$. Due to the flexibility of allowing for heavy tails, this assumption is still weaker than the counterpart light-tailed assumptions by Guigues et al. (2017).

Our most stringent condition on the underlying randomness is to require the cost function be sub-exponential for all $\mathbf{x} \in \mathcal{X}$, as formalized below.

**Assumption 3.7.** For any given $\mathbf{x} \in \mathcal{X}$, it holds that

$$\|f(\mathbf{x}, \xi) - F(\mathbf{x})\|_{\psi_1} \leq \sigma_\psi,$$

for some $\sigma_\psi \geq 0$.

*Remark* 3.8. In Assumption 3.7 we focused on light-tailed random variables, or more precisely sub-exponential random variables with their finite sub-exponential norm. Here the sub-exponential norm of a random variable $X$, denoted $\|X\|_{\psi_1}$, is defined as $\inf\{t > 0 : \mathbb{E}[\exp(|X|/t)] \leq 2\}$. This is the condition comparable to the counterpart assumption in the benchmark results in the literature, including those by Guigues et al. (2017); Lan et al. (2012).

## 4. Non-Asymptotic Confidence Intervals

In this section, we present our promised non-asymptotic confidence intervals for convex and nonconvex cases in Sections 4.1 and 4.2, respectively, and some discussions on the evaluation of important quantities in our result in Sections 4.3.

### 4.1. SAA-Based NCBs for Convex SO

This section presents (in Theorem 4.1) our results on constructing computable NCBs using information from the conventional SAA formulation.

**Theorem 4.1** (Convex SO, heavy-tailed). *Let $F(\cdot)$ be differentiable at $\mathbf{x}^*$ and $f(\cdot, \xi)$ be convex over a neighborhood of $\mathcal{X}$ for all $\xi \in \Xi$. Assume that the $q$-norm (where $q = p/(1-p)$) diameter of $\mathcal{X}$ is bounded from above by $\mathcal{D}_q$. Suppose that Assumptions 3.1 and 3.3 hold with some $p \geq 2$. For any $\alpha \in (0, 1)$, any $\delta$-optimal solution $\widehat{\mathbf{x}}_\delta$ to (3) satisfies the below with probability at least $1 - \alpha$*

$$F^* \in \left[ F_N(\widehat{\mathbf{x}}_\delta) - \frac{2.74}{\alpha^{1/p}} \cdot 6^{1/p} \cdot \sigma_f \cdot \sqrt{\frac{p}{N}} - \delta, \right.$$

$$\left. F_N(\widehat{\mathbf{x}}_\delta) + \frac{2.74}{\alpha^{1/p}} \cdot 6^{1/p} \cdot (\sigma_g \mathcal{D}_q + \sigma_f) \cdot \sqrt{\frac{p}{N}} \right]. \quad (10)$$

*Proof.* See Section A ☐

As mentioned in Remarks 3.4, our assumptions on underlying randomness in Theorem 4.1 are weaker than the benchmark results by Guigues et al. (2017). Under these conditions, the said theorem is perhaps the first NCB result for convex SO in non-Lipschitzian and heavy-tailed settings. In particular, our heavy-tailed assumptions are only imposed locally at $\mathbf{x}^*$ alone, allowing for further flexibility. As a quality measure for confidence intervals, the length of the confidence interval in (10) shrinks in terms of sample size with $O(N^{-1/2})$, matching with the result in Guigues et al. (2017).

*Remark* 4.2. Our results in Theorem 4.1 are non-Lipschitzian; namely, NCBs do not increase with a larger Lipschitz constant when all other quantities are fixed.

The metric entropy terms, such as the logarithm of the covering number, may elevate the NCBs' dependence on problem dimensionality, crucial consideration for larger-scale problems. Our results in Theorem 4.1 do not depend on any quantification of metric entropy. This feature, combined with Remark 4.2, perhaps often makes the NCB results sharper than the metric entropy-dependent results (Oliveira & Thompson, 2023).

*Remark* 4.3. Theorem 4.1 describes how the confidence interval changes with respect to the highest order of existing moments $p$, and thus explains how the convergence rate of

confidence bounds can be strengthened with lighter tails. Particularly, our result is also comparable with light-tailed results, as we specify $p = \ln(6/\alpha)$ as the following:

$$L(N, \alpha) := 2.74 \cdot e \cdot \sigma_f \cdot \sqrt{\frac{\ln(6/\alpha)}{N}} - \delta,$$

$$U(N, \alpha) := 2.74 \cdot e \cdot (\sigma_g \mathcal{D}_q + \sigma_f) \cdot \sqrt{\frac{\ln(6/\alpha)}{N}}.$$

*Remark* 4.4. The constant 2.74 in (10) originates from Marcinkiewicz's inequality, as is shown in Lemma D. The estimation of the universal constant in Marcinkiewicz's inequality was provided in Theorem 15.4 in Boucheron et al. (2013), where the constant is shown to satisfy the bound $K < 0.935$. With elementary level of algebra the constant propagates to 2.74 in (10).

*Remark* 4.5. The factor $\sqrt{p}$ in (10) also originates from Marcinkiewicz's inequality. Notably, Theorem 4.1 holds for all admissible $p$ given the problem assumption, and yet not necessary to be the highest order of existence moments. One may choose $p$ from all possible values and select the one that would lead to the smallest length of the confidence interval.

### 4.2. DRM-Based NCBs for Nonconvex SO

This subsection presents our results on nonconvex SO. In nonconvex light-tailed case, our non-Lipschitzian result is divided into two parts: (i) Lemma 4.6 considers the SAA problem in (3) to provide a lower bound for $F^*$; and (ii) Theorem 4.7 considers the DRM problem in (4) to provide an upper bound for $F^*$. We further extend these results to heavy tailed problems and show our DRM-based NCBs for over-parameterized models in Theorem 4.13.

Our light-tailed assumption (Assumption 3.7) leads to the well-known Bernstein's inequality (provided, e.g., by Vershynin (2018) and Zhang & Songxi (2021) with implicit and explicit universal constants, respectively): for any $t \geq 0$ and fixed $\mathbf{x} \in \mathcal{X}$,

$$\text{Prob}\left[\left|\frac{1}{N}\sum_{j=1}^{N} f(\mathbf{x}, \xi_j) - F(\mathbf{x})\right| \geq 4\sigma_\psi \cdot \left(\sqrt{\frac{t}{2N}} + \frac{t}{N}\right)\right] \leq 2\exp(-t). \quad (11)$$

**Lemma 4.6** (Nonconvex SO, light-tailed, lower bound). *Recall the definition of $F_N^*$ in (3). Under Assumption 3.7, it holds that*

$$F^* \geq F_N^* - 4\sigma_\psi\sqrt{\frac{1}{2N}} \cdot \sqrt{\ln\frac{4}{\alpha}} + \frac{4\sigma_\psi}{N} \cdot \left(\ln\frac{4}{\alpha}\right)$$

*with probability at least $1 - \alpha/2$, for any $\alpha \in (0, 1)$.*

*Proof.* With the fact that

$$F_N^* \leq N^{-1}\sum_{j=1}^{N} f(\mathbf{x}^*, \xi_j),$$

the proof is done by letting $t = \ln(4/\alpha)$ in (11). □

The lower bound for the result is straightforward. On the other hand, the upper bound becomes more critical and challenging to achieve a non-Lipschitzian result. By examining the DRM problem in (4) under nonconvexity and light-tailedness, Theorem 4.7 builds the promised upper bound of confidence interval.

**Theorem 4.7** (Nonconvex SO, light-tailed, upper bound). *Let $\widehat{\mathbf{x}}_{\delta,\gamma}$ be a $\delta$-optimal solution to (4). Under Assumption 3.7, for any $\alpha \in (0, 1)$, it holds that*

$$F^* \leq \sup_{\|\Delta\mathbf{x}\| \leq \gamma} F_N(\widehat{\mathbf{x}}_{\delta,\gamma} + \Delta\mathbf{x})$$
$$+ 4\sigma_\psi\sqrt{\frac{1}{2N}} \cdot \sqrt{\frac{\Delta_{p,q}d^{2/q}}{\gamma^2} + \ln\frac{4}{\alpha}}$$
$$- \frac{4\sigma_\psi}{N} \cdot \left(\frac{\Delta_{p,q}d^{2/q}}{\gamma^2} + \ln\frac{4}{\alpha}\right) \quad (12)$$

*with probability at least $1 - \alpha/2$, where*

$$\Delta_{p,q} := \frac{4\pi\ln 2}{\pi^{1/q}} \cdot \left[\Gamma\left(\frac{q+1}{2}\right)\right]^{2/q} \cdot \max_{\mathbf{x}\in\mathcal{X}}\|\mathbf{x}\|_p^2,$$

*given any $(p, q)$ such that $p^{-1} + q^{-1} = 1$ and $p \geq 1$.*

*Proof.* See Section B □

In nonconvex cases, Theorem 4.7 constructs perhaps the first non-Lipschitzian upper bound for $F^*$. For light-tailed nonconvex settings, our NCBs are presented by Theorem 4.7 together with Lemma 4.6. Similar as what is stated in Remark 4.2, our results in Theorem 4.7 and Lemma 4.6 are non-Lipschitzian.

*Remark* 4.8. Theorem 4.7 is not free from metric entropy, as $\Delta_{p,q}$ can grow with dimensionality $d$. However, comparing with the comparable quantities in (9), $\Delta_{p,q}$ in (12) is much easier to estimate. By Stirling's approximation to factorials, we could specify $\Delta_{p,q}$ as the following:

$$\Delta_{p,q} \approx \left(\frac{1}{2e}\right)^{\frac{q+1}{q}} \cdot (q+1)^{\frac{q+2}{q}} \cdot \max_{\mathbf{x}\in\mathcal{X}}\|\mathbf{x}\|_p^2 \cdot 4\pi\ln 2,$$

then

$$\lim_{q\to\infty}\frac{\Delta_{p,q}}{q} = \frac{2\pi\ln 2}{e} \cdot \max_{\mathbf{x}\in\mathcal{X}}\|\mathbf{x}\|_p^2. \quad (13)$$

This constant limitation in (13) shows that $\Delta_{p,q}$ grows almost linearly in $q$; namely, $\Delta_{p,q} \leq Cq \cdot \max_{\mathbf{x}\in\mathcal{X}}\|\mathbf{x}\|_p^2$ for some universal constant $C > 0$.

*Remark* 4.9. With the assumption that the problem scope is restricted to a bounded $1-$norm feasible set, the term $\|\mathbf{x}\|_p^2$ is bounded by some universal constant. If we further let $q = \ln d$, the dependence of dimensionality $d$ is thus logarithmic in Theorem 4.7.

*Remark* 4.10. The combination of Theorem 4.7 and Lemma 4.6 provides the desired NCBs in (6).

For heavy-tailed case, we impose a structural assumption as the below:

**Assumption 4.11.** The cost function is non-negative; namely, $f(\mathbf{x}, \xi) \geq 0$ for all $(\mathbf{x}, \xi) \in \mathcal{X} \times \Xi$ and the optimal solution $\widehat{\mathbf{x}}_\gamma$ to DRM (4) yields a zero objective function value, namely

$$\sup_{\|\Delta \mathbf{x}\| \leq \gamma} F_N(\widehat{\mathbf{x}}_\gamma + \Delta \mathbf{x}) = 0.$$

There are many scenarios in machine learning field where Assumption 4.11 is aligned with truth, e.g. mean squared error loss for single-label binary classification with no label ambiguity and there is non-zero margin of separation. Ideal choices of $\gamma$ then should be consistent with the said margin in order to preserve Assumption 4.11.

*Remark* 4.12. Assumption 4.11 is directly related to the assumption of the presence of a positive margin in separating the data population in the application of classification, such as Assumption 3.3 in Cao & Gu (2020). In this example, the assumed constant margin therein can imply the said perturbed settings.

**Theorem 4.13** (Over-parameterized models, nonconvex SO, heavy-tailed, upper bound)**.** *Suppose that Assumptions 4.11 and 3.5 hold. Let $\widehat{\mathbf{x}}_\gamma$ be an optimal solution to (4). Then for any $\alpha \in (0, 1)$, the inequality*

$$\sup_{\|\Delta \mathbf{x}\| \leq \gamma} F_N(\widehat{\mathbf{x}}_\gamma + \Delta \mathbf{x})$$

$$\geq F(\mathbf{x}^*) \cdot \left(1 - \frac{7}{\ln 2} \cdot \left(\frac{\Delta_{p,q} d^{2/q}}{\gamma^2} + \ln \frac{4}{\alpha}\right) \cdot N^{-\frac{1}{2}}\right)$$

$$- \frac{8}{\ln 2} \cdot \left(\frac{\Delta_{p,q} d^{2/q}}{\gamma^2} + \ln \frac{4}{\alpha}\right) \cdot \left(1 + \frac{\sigma_f'}{p-1}\right) \cdot N^{\frac{2-p}{2p}}$$

$$\tag{14}$$

*holds with probability at least $1 - \alpha/2$, where*

$$\Delta_{p,q} := \frac{4\pi \ln 2}{\pi^{1/q}} \cdot \left[\Gamma\left(\frac{q+1}{2}\right)\right]^{2/q} \cdot \max_{\mathbf{x} \in \mathcal{X}} \|\mathbf{x}\|_p^2,$$

*given any $(p, q)$ such that $p^{-1} + q^{-1} = 1$ and $p \geq 1$.*

*Proof.* See Section C $\qquad\square$

*Remark* 4.14. With the scope fixed for over-parameterized models, the light-tailed result in Theorem 4.7 could be extended to a heavy-tailed result, as is revealed by Theorem 4.13. Mathematically, this transition from light tail to heavy tail is achieved by the construction of a bounded auxiliary problem, which behaves equivalently to the heavy-tailed DRM under Assumption 4.11. See the proof in Section C for more details.

Invoking Markov's inequality under the heavy-tailed Assumption 3.3, we have

$$\text{Prob}\left[|F_N(\mathbf{x}^*) - F(\mathbf{x}^*)| \leq \frac{2^{1/p}}{\alpha^{1/p}} \|F_N(\mathbf{x}^*) - F(\mathbf{x}^*)\|_{L_p}\right]$$

$$\geq 1 - \alpha/2.$$

This together with Lemma D.1 then immediately leads to

$$\text{Prob}\left[|F_N(\mathbf{x}^*) - F(\mathbf{x}^*)| \leq \frac{2.74}{\alpha^{1/p}} \cdot 4^{1/p} \cdot \sigma_f \cdot \sqrt{\frac{p}{N}}\right]$$

$$\geq 1 - \alpha/2,$$

which then easily provides an lower bound on $F^*$. This lower bound of $F^*$, combined with Theorem 4.13, leads to the result in (7) as promised.

The confidence bounds inequality in (14) could be transformed into the format in (2) by elementary level of algebra, i.e.

$$U_{N,\alpha} = \left[F_\gamma^* + \frac{8}{\ln 2} \cdot \left(\frac{\Delta_{p,q} d^{2/q}}{\gamma^2} + \ln \frac{4}{\alpha}\right)\right.$$

$$\left. \cdot \left(1 + \frac{\sigma_f'}{p-1}\right) \cdot N^{-\frac{p-2}{2p}}\right]$$

$$\cdot \left(1 - \frac{7}{\ln 2} \cdot \left(\frac{\Delta_{p,q} d^{2/q}}{\gamma^2} + \ln \frac{4}{\alpha}\right) \cdot N^{-\frac{1}{2}}\right)^{-1},$$

which is shown earlier in (7). As mentioned in Remark 4.8, $\Delta_{p,q}$ also grows almost linearly in $q$ as in Theorem 4.13.

*Remark* 4.15. Similar as what is stated in Remark 4.2, our results for nonconvex heavy-tailed NCB are still non-Lipschitzian.

### 4.3. Discussions on $F_N^*$ and $F_\gamma^*$

In the construction of NCBs, quantities such as $F_N^*$ and $F_\gamma^*$ need to be carefully evaluated to ensure they are computable for constructing NCBs. The pseudo-code for evaluating these quantities is summarized below:

---

**Algorithm 1** Evaluation of $F_N^*$ and $F_\gamma^*$

---

**Input:** i.i.d. sampled data points $\mathbf{p}_i \in \mathbb{R}^m$, with sample size $N$

**Solve for**

$F_N^* = \inf_{\mathbf{x} \in \mathcal{X}} N^{-1} \sum_{i=1}^N f(\mathbf{x}, \mathbf{p}_i)$

$F_\gamma^* = \inf_{\mathbf{x} \in \mathcal{X}} \sup_{\|\mathbf{v}\| \leq \gamma} N^{-1} \sum_{i=1}^N f(\mathbf{x} + \mathbf{v}, \mathbf{p}_i)$

**Output:** $F_N^*, F_\gamma^*$

---

Many studies provide alternatives for solving $F_N^*$ and $F_\gamma^*$ (as per, e.g., Shapiro et al., 2021; Kleywegt et al., 2002; Birge & Louveaux, 2011; Nemirovski et al., 2009; Norton & Royset, 2023; Zheng et al., 2020), and the solutions to SAA problem are much more frequently visited. Algorithms for solving SAA, like subgradient descent, have been tested both for their strong numerical performance and an optimal (or sometimes near optimal) theoretical convergence rate. As a robust modification of SAA, solving DRM is generally more expensive than SAA. In general cases, most of the existing procedure on subgradient method could be directly applied to DRM, but at the cost of estimating the true subgradient for DRM formulation. The computation of such subgradient usually incorporates the computation over a large number of points within the $\gamma$-radius ball centered at the current solution. As a result, the key element in DRM algorithm is to build efficient and effective approximations to the true subgradient of DRM. With well-structured approximation design, DRM can take as approximately $3 - 5$ times longer than SAA over the same architecture and dataset, as noted in Norton & Royset (2023).

## 5. Numerical Experiments

We conducted two sets of experiments to evaluate our results as reported in the sequel. The first experiment was performed on a stochastic linear programming problem. The second experiment was focused on the training formulation of an over-parameterized neural network.

### 5.1. Convex NCBs in Stochastic Linear Problem

Our first experiment concern the stochastic linear program formulation: $\min \left\{ \mathbb{E}[f(\mathbf{x}, \xi)] : \mathbf{x} \geq 0, \mathbf{1}^\top \mathbf{x} = 1 \right\}$, where $\mathbf{1}$ is an all-one vector and $f(\mathbf{x}, \xi) = -\sum_{i=1}^d \kappa_i \xi_i x_i$. Here, $\kappa_i = 0.08 + 0.04(i - 1)/d$, for $i = 1, \ldots, d$ and $\xi_1, \xi_2, \ldots, \xi_d$ are i.i.d. copies of $\xi$, a power-law distributed random variable with probability density function $p_\xi(x) = ab^a/x^{a+1}$ for $x > b$ with distribution parameters $a$ and $b$. One may verify that the highest order of existence of moments is $a - 1$. We specified $a = 3.01, b = 1$ (correspondingly $p = 2$), and the significance level to be $\alpha = 0.01$. Other problem quantities in the NCBs were estimated with approaches discussed in Appendix E.1. The optimal solutions to both the SAA and the exact SO problems admit straightforward closed forms. We calculated the empirical coverage probability (ECP) — the proportion of replications in which $F^*$ lies within the calculated NCBs $[L_{N,\alpha}, U_{N,\alpha}]$ — out of 10,000 independent random replications. The results, as presented in Table 3 of Appendix E.3, show that the proposed NCBs can achieve high ECPs comparable to some benchmark scheme derived by Oliveira & Thompson (2023).

We further evaluated the length of the NCBs relative to the existing benchmarks. To that end, we calculated the ratio $r_1$ between the length of our proposed NCBs (referred to as P-NCBs below) and that of the benchmark NCBs (refered to as B-NCBs), defined as

$$r_1 := \frac{U_{\text{P-NCB}} - L_{\text{P-NCB}}}{U_{\text{B-NCB}} - L_{\text{B-NCB}}}, \tag{15}$$

where $U_{\text{P-NCB}}$ and $L_{\text{P-NCB}}$ denote the upper and lower bounds constructed by the proposed NCBs as in (5). $U_{\text{B-NCB}}$ and $L_{\text{B-NCB}}$ denote the benchmark upper and lower bounds provided by Theorem 3 in Oliveira & Thompson (2023). For schemes with comparable ECP, smaller $r_1$ ratios indicate narrower, and thus more desirable, NCBs.

The results are presented in both Table 1 and Figure 1, where we distinguish between two variants of P-NCB:

- P-NCB$^E$: the proposed NCBs constructed with estimated problem quantities to mimic practical settings with limited prior knowledge (refer to Appendix E.1 for details);

- P-NCB$^*$: the proposed NCBs constructed with exact problem quantities as in idealized cases.

*Table 1.* Length ratio $r_1$ generated by P-NCB$^E$ and P-NCB$^*$ relative to the benchmark for convex SO problems with different problem dimensionality $d$

| Method | $d$ | | | | |
|---|---|---|---|---|---|
| | 100 | 500 | 1000 | 2000 | 4000 |
| P-NCB$^E$ | 0.599 | 0.419 | 0.300 | 0.215 | 0.170 |
| P-NCB$^*$ | 0.281 | 0.170 | 0.120 | 0.085 | 0.064 |

As shown in both Table 1 and Figure 1, the length ratios $r_1$ are less than 1 across different problem dimensionalities. This indicates that both P-NCB$^E$ and P-NCB$^*$ produced narrower and thus sharper confidence bounds than the benchmark scheme B-NCB. Furthermore, in between P-NCB$^E$ and P-NCB$^*$, the latter is noticeably more preferable when the true values of problem quantities can be accessible.

In addition, both P-NCB$^E$ and P-NCB$^*$ exhibit a monotonic decrease in $r_1$ as $d$ grows. This decreasing trend shows the scalability of P-NCB to higher-dimensional problems, as the practical benefit of eliminating the Lipschitz constants in our new derivations.

Note that, given the same set of problem quantities (such as $d$, $\sigma_f$, $\sigma_g$, $\sigma_\psi$, and $\mathcal{D}_q$), the length ratio remains the same for different choices of sample size $N$. This is because that both P-NCB and B-NCB depends on $N$ at the same rate of $O(1/\sqrt{N})$, which is canceled out in calculating the length ratio in (15).

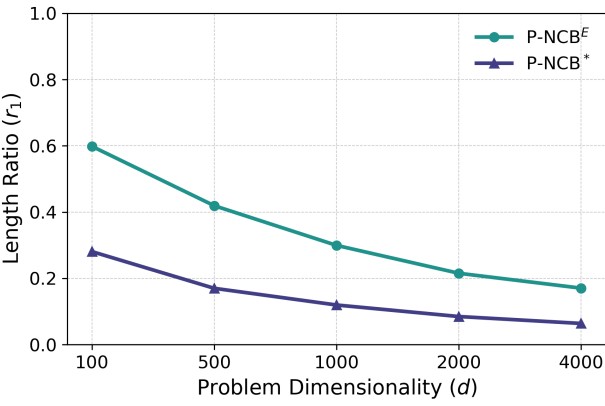

*Figure 1.* The length ratios $r_1$ generated by P-NCB$^E$ and P-NCB$^*$ relative to the benchmark across increasing problem dimensionality $d$. The x-axis shows dimensionality (from 100 to 4000), and the y-axis shows the length ratios. Lower values correspond to tighter confidence bounds provided by P-NCB, comparing with B-NCB.

### 5.2. Nonconvex NCBs in Over-Parameterized Model

For nonconvex problem, we considered the problem of training a two-layer neural network with a data generating process $Y = h(X) + \epsilon$, where $h(\cdot)$ is some unknown function in nature to be reconstructed through the observations of some $N$-many sample points of $(X, Y)$. For the purpose of simulation, we specified $h(\cdot)$ to be a two-layer neural network with LeakyRelu activation function and randomly simulated weights. Here, $\epsilon$ is assumed to follow student $t$-distribution to simulate heavy-tailed underlying randomness. We solved the SAA and DRM problem formulations corresponding to training the said neural network in minimizing mean squared error with different combinations of the sample size $N$ and problem dimensionality $d$.

Under significance level $\alpha = 0.1$, we present the average length ratio $r_2$ between our proposed nonconvex NCBs (referred to as P-NCB-NC$^E$) and that of the nonconvex benchmark NCBs (referred to as B-NCB-NC$^E$) in Table 2, defined as

$$r_2 := \frac{1}{10} \cdot \sum_{m=1}^{10} \frac{(U_{\text{P-NCB-NC}}^m - L_{\text{P-NCB-NC}}^m)}{U_{\text{B-NCB-NC}}^m - L_{\text{B-NCB-NC}}^m},$$

where $U_{\text{P-NCB-NC}}^m$ and $L_{\text{P-NCB-NC}}^m$ denote the upper and lower bounds constructed by the proposed nonconvex NCBs as in (7) for one randomly generated problem instance. For each problem instance, the problem quantities were estimated to mimic practical settings with limited prior knowledge. We repeat the calculation for 10 times. $U_{\text{B-NCB-NC}}^m$ and $L_{\text{B-NCB-NC}}^m$ denote the corresponding nonconvex upper and lower bounds provided by a benchmark scheme (as derived in Theorem 2 of Oliveira & Thompson (2023)).

Note here, in those replications, the nonconvexity of the formulation led to different solutions in computing the SAA and DRM formulations, which further resulted in different NCBs. We present in Table 2 the average length ratio $r_2$ of

*Table 2.* Average length ratio $r_2$ generated by P-NCB-NC$^E$ relative to B-NCB-NC$^E$ for nonconvex SO

| $N$ | $d = 41$ | $N$ | $d = 961$ | $N$ | $d = 1681$ |
|---|---|---|---|---|---|
| 300 | 0.030 | 500 | 0.023 | 500 | 0.009 |
| 340 | 0.018 | 600 | 0.012 | 600 | 0.009 |
| 380 | 0.018 | 700 | 0.012 | 700 | 0.009 |
| 420 | 0.017 | 800 | 0.011 | 800 | 0.008 |

P-NCB-NC$^E$. As shown in Table 2, an average ratio below one indicates that P-NCB-NC$^E$ produces smaller lengths, resulting in tighter bounds.

The average length ratio $r_2$ decreases consistently as both the sample size $N$ and problem dimensionality $d$ increase, and thus confirms the relative improvement of the proposed NCBs. Such a consistent reduction highlights the scalability of our approach to higher-dimensional problems due to the non-Lipschitzian feature of our results.

We further tested the ECP of both methods, which showed comparable results. We present these results in Table 4 of Appendix E.1.

## 6. Conclusions

This paper revisits SAA and its robust reformulation DRM, and constructs the confidence interval (i.e., both upper and lower confidence bounds) for the estimation of the true objective function value. Motivated by the sensitivity to the Lipschitz constant in the existing non-asymptotic confidence interval results, this paper presents the non-Lipschitzan results under three sets of assumptions (i) the convex SO problems under heavy-tailedness; (ii) the noncovex SO problems under light-tailedness; and (iii) the nonconvex over-parameterized models under heavy-tailedness. Our results show, for the first time, that the elimination of Lipschitz constant (in all cases) as well as metric entropy (under heavy-tailedness and convexity) is achievable, resulting in a reduction in the dependence on both Lipschitz conditions and problem dimensionality for the construction of non-asymptotic confidence intervals for SO. Our numerical experiments showed an improvement of our method relative to some existing benchmark especially for higher-dimensional cases. However, some level of conservatism still remains in the constructed bounds, which we will seek further improved in future research.

## Acknowledgements

The authors would like to thank the anonymous reviewers for their active participation in the review-rebuttal cycle as well as their insightful and constructive comments. Liu and Tong are partially supported by NSF CMMI 2213459. Royset is supported in part by Office of Naval Research under Grant N00014-24-1-2492.

## Impact Statement

This paper presents work whose goal is to advance the field of Machine Learning. There are many potential societal consequences of our work, none which we feel must be specifically highlighted here.

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

## A. Proof of Theorem 4.1

*Proof.* **Step 1.** Invoking Markov's inequality under Assumption 3.3, we have

$$\text{Prob}\left[|F_N(\mathbf{x}^*) - F(\mathbf{x}^*)| \leq \frac{3^{1/p}}{\alpha^{1/p}} \cdot \|F_N(\mathbf{x}^*) - F(\mathbf{x}^*)\|_{L_p}\right] \geq 1 - \alpha/3.$$

Combining this with Lemma D.1, we immediately have

$$\text{Prob}\left[|F_N(\mathbf{x}^*) - F(\mathbf{x}^*)| \leq \frac{2.74}{\alpha^{1/p}} \cdot 6^{1/p} \cdot \sigma_f \cdot \sqrt{\frac{p}{N}}\right] \geq 1 - \alpha/3. \tag{16}$$

By definition, $F_N(\widehat{\mathbf{x}}_\delta) \leq F_N(\mathbf{x}^*) + \delta$. Therefore, $F_N(\widehat{\mathbf{x}}_\delta) - F(\mathbf{x}^*) \leq F_N(\mathbf{x}^*) - F(\mathbf{x}^*) + \delta$. This combined with (16) leads to

$$\text{Prob}\left[F_N(\widehat{\mathbf{x}}_\delta) - F(\mathbf{x}^*) \leq \frac{2.74}{\alpha^{1/p}} \cdot 6^{1/p} \cdot \sigma_f \cdot \sqrt{\frac{p}{N}} + \delta\right] \geq 1 - \alpha/3. \tag{17}$$

**Step 2.** Let $\partial F_N(\mathbf{x}^*)$ be subdifferential of $F_N$ at $\mathbf{x}^*$. By Markov's inequality and the differentiability of $F$ at $\mathbf{x}^*$, if we also let $\mathbf{g}_N^* = N^{-1}\sum_{j=1}^N g_f^*(\xi_j)$ with $g_f^*(\xi) \in \partial f(\mathbf{x}^*, \xi)$ defined as in Assumption 3.1, then

$$\text{Prob}\left[\|\mathbf{g}_N^* - \nabla F(\mathbf{x}^*)\|_p \leq \frac{3^{1/p}}{\alpha^{1/p}} \|\mathbf{g}_N^* - \nabla F(\mathbf{x}^*)\|_{L_p}\right] \geq 1 - \frac{\alpha}{3},$$

which, combined with invoking lemma D.1 under Assumption 3.1, implies that

$$\text{Prob}\left[\|\mathbf{g}_N^* - \nabla F(\mathbf{x}^*)\|_p \leq \frac{2.74}{\alpha^{1/p}} \cdot 6^{1/p} \cdot \sigma_g \cdot \sqrt{\frac{p}{N}}\right] \geq 1 - \alpha/3. \tag{18}$$

Now, observe that, by convexity of $f(\cdot, \xi)$ for all $\xi \in \Xi$,

$$\begin{aligned}
F_N(\widehat{\mathbf{x}}_\delta) - F_N(\mathbf{x}^*) &\geq \langle \mathbf{g}_N^*, \widehat{\mathbf{x}}_\delta - \mathbf{x}^* \rangle \\
&= \langle \mathbf{g}_N^* - \nabla F(\mathbf{x}^*), \widehat{\mathbf{x}}_\delta - \mathbf{x}^* \rangle + \langle \nabla F(\mathbf{x}^*), \widehat{\mathbf{x}}_\delta - \mathbf{x}^* \rangle \\
&\geq \langle \mathbf{g}_N^* - \nabla F(\mathbf{x}^*), \widehat{\mathbf{x}}_\delta - \mathbf{x}^* \rangle \geq -\|\mathbf{g}_N^* - \nabla F(\mathbf{x}^*)\|_p \cdot \mathcal{D}_q,
\end{aligned}$$

which implies that

$$F_N(\widehat{\mathbf{x}}_\delta) - F(\mathbf{x}^*) \geq F_N(\mathbf{x}^*) - F(\mathbf{x}^*) - \|\mathbf{g}_N^* - \nabla F(\mathbf{x}^*)\|_p \cdot \mathcal{D}_q.$$

Combining this with (16) and (18) through the union bound implies that

$$\text{Prob}\left[F_N(\widehat{\mathbf{x}}_\delta) - F(\mathbf{x}^*) \geq -\frac{2.74}{\alpha^{1/p}} \cdot 6^{1/p} \cdot (\sigma_g \cdot \mathcal{D}_q + \sigma_f) \cdot \sqrt{\frac{p}{N}}\right] \geq 1 - 2\alpha/3. \tag{19}$$

Invoking the union bound again in joining (17) and (19) then leads to the desired result. $\qquad\square$

## B. Proof of Theorem 4.7

*Proof.* By the argument of $\epsilon$-net, there exists a sequence of solutions $\mathcal{V}_K := \{\mathbf{x}_1, ..., \mathbf{x}_K\} \subseteq \mathcal{X}$ such that $\max_{\mathbf{x}\in\mathcal{X}} \min_{\mathbf{y}\in\mathcal{V}_K} \|\mathbf{x} - \mathbf{y}\| \leq \gamma$ and that $K := |\mathcal{V}_K|$ satisfies the following inequality according to Sudakov's minoration inequality (as presented in Theorem 7.4.1 by Vershynin (2018), where we can explicate the constant therein to $c := \frac{1}{\sqrt{2\pi \cdot \ln 2}}$ by invoking the fact that the expected number of $K$-many standard normal random variables is at least $\frac{1}{\sqrt{\pi \cdot \ln 2}} \cdot \sqrt{\ln K}$ as per Kamath (2015)):

$$\frac{\gamma}{\sqrt{2\pi \ln 2}} \sqrt{\ln K} \leq \mathbb{E}\left[\sup_{\mathbf{x}\in\mathcal{X}} \langle \mathbf{g}, \mathbf{x} \rangle\right], \tag{20}$$

for a vector $\mathbf{g}$ of standard Gaussian random variables. Note it holds that $\mathbb{E}\left[\sup_{\mathbf{x}\in\mathcal{X}}\langle\mathbf{g},\mathbf{x}\rangle\right] \leq \max_{\mathbf{x}\in\mathcal{X}}\|\mathbf{x}\|_p \cdot \mathbb{E}[\|\mathbf{g}\|_q] \leq \max_{\mathbf{x}\in\mathcal{X}}\|\mathbf{x}\|_p \cdot (\mathbb{E}[\|\mathbf{g}\|_q^q])^{1/q}$, where the last inequality is due to the concavity of $(\cdot)^c$ on $\mathbb{R}_+$ for any $c \in (0,1)$. Since the $i$th component of $\mathbf{g}$, denoted by $g_i$, satisfies that $\mathbb{E}|g_i|^q = \frac{1}{\sqrt{\pi}} \cdot 2^{q/2} \cdot \Gamma(\frac{q+1}{2})$. Then, $\mathbb{E}[\|\mathbf{g}\|_q^q] = \sum_{i=1}^d \mathbb{E}[|g_i|^q] = \frac{d}{\sqrt{\pi}} \cdot 2^{q/2} \cdot \Gamma(\frac{q+1}{2})$. This combined with (20) then implies that

$$\ln K \leq \frac{4\pi\ln 2}{\gamma^2} \cdot \frac{d^{2/q}}{\pi^{1/q}} \cdot \left[\Gamma\left(\frac{q+1}{2}\right)\right]^{2/q} \cdot \max_{\mathbf{x}\in\mathcal{X}}\|\mathbf{x}\|_p^2 =: \frac{\Delta_{p,q}d^{2/q}}{\gamma^2}, \tag{21}$$

where $\Delta_{p,q} := \frac{4\pi\ln 2}{\pi^{1/q}} \cdot \left[\Gamma\left(\frac{q+1}{2}\right)\right]^{2/q} \cdot \max_{\mathbf{x}\in\mathcal{X}}\|\mathbf{x}\|_p^2$.

Under Assumption 3.7 (which implies (11)) and invoking the union bound and De Morgan's law, we then have

$$\text{Prob}\left[\max_{k=1,\dots,K}\left|N^{-1}\sum_{j=1}^N f(\mathbf{x}_k,\xi_j) - F(\mathbf{x}_k)\right| \leq 4\sigma_\psi \cdot \left(\sqrt{\frac{t}{2N}} + \frac{t}{N}\right)\right]$$
$$\geq 1 - 2K \cdot \exp(-t) \geq 1 - 2\exp\left(\frac{\Delta_{p,q}d^{2/q}}{\gamma^2} - t\right),$$

where the last inequality is the result of (21). Since $F(\mathbf{x}_k) \geq F(\mathbf{x}^*)$ by the definition of $\mathbf{x}^*$, we have that

$$F_N(\mathbf{x}_k) \geq F(\mathbf{x}^*) - 4\sigma_\psi \cdot \left(\sqrt{\frac{t}{2N}} + \frac{t}{N}\right), \quad \forall k = 1,\dots,K,$$

with probability at least $1 - 2\exp\left(\frac{\Delta_{p,q}d^{2/q}}{\gamma^2} - t\right)$. By the construction of this $\epsilon$-net, we know that $\|\widehat{\mathbf{x}}_{\delta,\gamma} - \mathbf{x}_\kappa\| \leq \gamma$ for some $\kappa = 1,\dots,K$. Consequently,

$$\sup_{\|\Delta\mathbf{x}\|\leq\gamma} F_N(\widehat{\mathbf{x}}_{\delta,\gamma} + \Delta\mathbf{x}) \geq F_N(\mathbf{x}_\kappa) \geq F(\mathbf{x}^*) - 4\sigma_\psi \cdot \left(\sqrt{\frac{t}{2N}} + \frac{t}{N}\right)$$

with probability at least $1 - 2\exp\left(\frac{\Delta_{p,q}d^{2/q}}{\gamma^2} - t\right)$. By letting $t = \frac{\Delta_{p,q}d^{2/q}}{\gamma^2} + \ln(\frac{4}{\alpha})$, we immediately have the desired result. $\qquad\square$

## C. Proof of Theorem 4.13

*Proof.* To start the formal proof of Theorem 4.13, we begin with a corollary extending the result in Theorem 4.7 to bounded random variables. Later one would see how this bounded settings could strengthen our over-parameterized models result.

**Corollary C.1** (Bounded random variable). *Suppose $f(\mathbf{x},\xi)$ is bounded with $L \leq f(\mathbf{x},\xi) \leq U$, and $\widehat{\mathbf{x}}_{\delta,\gamma}$ is a $\delta$-optimal solution to* (4) *satisfying all the conditions in Theorem 4.7, it holds that*

$$\sup_{\|\Delta\mathbf{x}\|\leq\gamma} F_N(\widehat{\mathbf{x}}_{\delta,\gamma} + \Delta\mathbf{x}) \geq F(\mathbf{x}^*) - \frac{4(U-L)}{\ln 2} \cdot \sqrt{\frac{1}{2N}} \cdot \sqrt{\frac{\Delta_{p,q}d^{2/q}}{\gamma^2} + \ln\frac{4}{\alpha}} - \frac{4(U-L)}{\ln 2} \cdot \frac{1}{N} \cdot \left(\frac{\Delta_{p,q}d^{2/q}}{\gamma^2} + \ln\frac{4}{\alpha}\right)$$

*with probability at least $1 - \alpha/2$.*

The proof of Corollary C.1 is trivial. Note that $L - U \leq f(\mathbf{x},\xi) - F(\mathbf{x}) \leq U - L$, then

$$\mathbb{E}\left[\exp\left(\frac{|f(\mathbf{x},\xi) - F(\mathbf{x})|}{(U-L)/\ln 2}\right)\right] \leq 2.$$

By the definition of sub-exponential norm, $\|f(\mathbf{x},\xi) - F(\mathbf{x})\|_{\psi_1} \leq \frac{U-L}{\ln 2}$. This combined with (12) finishes the proof of Corollary C.1.

With Corollary C.1 concludes the bounded setting, we could construct a bounded auxiliary problem:

$$\max_{\|\Delta\mathbf{x}\| \leq \gamma} \min_{\mathbf{x} \in \mathcal{X}} N^{-1} \sum_{j=1}^{N} \min\{f(\mathbf{x} + \Delta\mathbf{x}, \xi_j), \Gamma \cdot N\}. \tag{22}$$

By Assumption 4.11, it holds that $f(\widehat{\mathbf{x}}_\gamma, \xi_j) = 0$. Thus, for any $\Gamma \geq 0$, it holds that $\widehat{\mathbf{x}}_\gamma$ is also an optimal solution to (22). Let $h(\mathbf{x}, \xi) := \min\{f(\mathbf{x}, \xi), \Gamma \cdot N\}$, $\mathbf{x}_H^*$ is an optimal solution to $\min_{\mathbf{x} \in \mathcal{X}} H(\mathbf{x}) := \mathbb{E}[h(\mathbf{x}, \xi)]$ and $H_N(\mathbf{x}) := N^{-1} \sum_{j=1}^{N} \min\{f(\mathbf{x}, \xi_j), \Gamma \cdot N\}$. Note that $0 \leq \min\{f(\mathbf{x}, \xi), \Gamma \cdot N\} \leq \Gamma \cdot N$, by Corollary C.1 and the definition of $H_N(\cdot)$, we immediately have

$$\sup_{\|\Delta\mathbf{x}\| \leq \gamma} F_N(\widehat{\mathbf{x}}_\gamma + \Delta\mathbf{x}) \geq \sup_{\|\Delta\mathbf{x}\| \leq \gamma} H_N(\widehat{\mathbf{x}}_\gamma + \Delta\mathbf{x})$$

$$\geq H(\mathbf{x}_H^*) - \frac{4\Gamma \cdot N}{\ln 2} \cdot \sqrt{\frac{1}{2N}} \cdot \sqrt{\frac{\Delta_{p,q} d^{2/q}}{\gamma^2} + \ln \frac{4}{\alpha}} - \frac{4\Gamma \cdot N}{\ln 2} \cdot \frac{1}{N} \cdot \left(\frac{\Delta_{p,q} d^{2/q}}{\gamma^2} + \ln \frac{4}{\alpha}\right) \tag{23}$$

with probability at least $1 - \alpha/2$. Let $\Gamma = N^{(1-p)/p} + F(\mathbf{x}_H^*) \cdot N^{-1}$, we have

$$F(\mathbf{x}^*) - H(\mathbf{x}_H^*) \leq F(\mathbf{x}_H^*) - H(\mathbf{x}_H^*)$$

$$= \int_0^\infty \mathrm{Prob}[f(\mathbf{x}_H^*, \xi) > t]dt - \int_0^\infty \mathrm{Prob}[\min\{f(\mathbf{x}_H^*, \xi), \Gamma \cdot N\} > t]dt$$

$$= \int_0^\infty \mathrm{Prob}[f(\mathbf{x}_H^*, \xi) > t]dt - \int_0^{\Gamma \cdot N} \mathrm{Prob}[f(\mathbf{x}_H^*, \xi) > t]dt = \int_{\Gamma \cdot N}^\infty \mathrm{Prob}[f(\mathbf{x}_H^*, \xi) > t]dt$$

$$= \int_{\Gamma \cdot N}^\infty \mathrm{Prob}[f(\mathbf{x}_H^*, \xi) - F(\mathbf{x}_H^*) > t - F(\mathbf{x}_H^*)]dt$$

$$\leq \int_{\Gamma \cdot N}^\infty \mathrm{Prob}[|f(\mathbf{x}_H^*, \xi) - F(\mathbf{x}_H^*)| > t - F(\mathbf{x}_H^*)]dt$$

$$\leq \int_{\Gamma \cdot N}^\infty \frac{\|f(\mathbf{x}_H^*, \xi) - F(\mathbf{x}_H^*)\|_{L^p}}{(t - F(\mathbf{x}_H^*))^p} dt \tag{24}$$

$$\leq \int_{\Gamma \cdot N}^\infty \frac{\sigma_f'}{(t - F(\mathbf{x}_H^*))^p} dt \tag{25}$$

$$= \frac{\sigma_f'}{p-1} \cdot \frac{1}{(\Gamma \cdot N - F(\mathbf{x}_H^*))^{p-1}} = \frac{\sigma_f'}{p-1} \cdot N^{-\frac{p-1}{p}}, \tag{26}$$

where (24) is due to Markov's inequality and (25) is due to Assumption 3.5. By the definition of $\mathbf{x}^*$ and $\mathbf{x}_H^*$ and (26),

$$F(\mathbf{x}^*) \leq F(\mathbf{x}_H^*) \leq H(\mathbf{x}_H^*) + \frac{\sigma_f'}{p-1} \cdot N^{-\frac{p-1}{p}} \leq F(\mathbf{x}^*) + \frac{\sigma_f'}{p-1} \cdot N^{-\frac{p-1}{p}}, \tag{27}$$

then by the construction of $\Gamma$ and (27),

$$\Gamma \leq N^{\frac{1}{p}-1} + \left(F(\mathbf{x}^*) + \frac{\sigma_f'}{p-1} \cdot N^{-\frac{p-1}{p}}\right) N^{-1}. \tag{28}$$

Due to the non-negativity of $\Delta_{p,q}, d, \gamma$, we know

$$\sqrt{\frac{\Delta_{p,q} d^{2/q}}{\gamma^2} + \ln \frac{4}{\alpha}} \leq \frac{\Delta_{p,q} d^{2/q}}{\gamma^2} + \ln \frac{4}{\alpha}. \tag{29}$$

Combining (27), (28), (23) and (29), we have

$$\sup_{\|\Delta \mathbf{x}\| \leq \gamma} F_N(\widehat{\mathbf{x}}_\gamma + \Delta \mathbf{x}) \tag{30}$$

$$\geq F(\mathbf{x}^*) - \frac{2\sqrt{2}}{\ln 2} \cdot \sqrt{\frac{\Delta_{p,q} d^{2/q}}{\gamma^2} + \ln \frac{4}{\alpha}} \cdot N^{\frac{2-p}{2p}} - \frac{2\sqrt{2}}{\ln 2} \cdot \sqrt{\frac{\Delta_{p,q} d^{2/q}}{\gamma^2} + \ln \frac{4}{\alpha}} \cdot F(\mathbf{x}^*) \cdot N^{-\frac{1}{2}}$$

$$- \frac{2\sqrt{2}}{\ln 2} \cdot \sqrt{\frac{\Delta_{p,q} d^{2/q}}{\gamma^2} + \ln \frac{4}{\alpha}} \cdot \frac{\sigma'_f}{p-1} \cdot N^{\frac{2-3p}{2p}} - \frac{4}{\ln 2} \cdot \left( \frac{\Delta_{p,q} d^{2/q}}{\gamma^2} + \ln \frac{4}{\alpha} \right) \cdot N^{\frac{1-p}{p}}$$

$$- \frac{4}{\ln 2} \cdot \left( \frac{\Delta_{p,q} d^{2/q}}{\gamma^2} + \ln \frac{4}{\alpha} \right) \cdot F(\mathbf{x}^*) \cdot N^{-1} - \frac{4}{\ln 2} \cdot \left( \frac{\Delta_{p,q} d^{2/q}}{\gamma^2} + \ln \frac{4}{\alpha} \right) \cdot \frac{\sigma'_f}{p-1} \cdot N^{\frac{1-2p}{p}} - \frac{\sigma'_f}{p-1} \cdot N^{\frac{1-p}{p}} \tag{31}$$

$$= F(\mathbf{x}^*) \cdot \left( 1 - \frac{2\sqrt{2}}{\ln 2} \cdot \sqrt{\frac{\Delta_{p,q} d^{2/q}}{\gamma^2} + \ln \frac{4}{\alpha}} \cdot N^{-\frac{1}{2}} - \frac{4}{\ln 2} \cdot \left( \frac{\Delta_{p,q} d^{2/q}}{\gamma^2} + \ln \frac{4}{\alpha} \right) \cdot N^{-1} \right)$$

$$- \frac{2\sqrt{2}}{\ln 2} \cdot \sqrt{\frac{\Delta_{p,q} d^{2/q}}{\gamma^2} + \ln \frac{4}{\alpha}} \cdot N^{\frac{2-p}{2p}} - \frac{2\sqrt{2}}{\ln 2} \cdot \sqrt{\frac{\Delta_{p,q} d^{2/q}}{\gamma^2} + \ln \frac{4}{\alpha}} \cdot \frac{\sigma'_f}{p-1} \cdot N^{\frac{2-3p}{2p}}$$

$$- \frac{4}{\ln 2} \cdot \left( \frac{\Delta_{p,q} d^{2/q}}{\gamma^2} + \ln \frac{4}{\alpha} \right) \cdot N^{\frac{1-p}{p}} - \frac{4}{\ln 2} \cdot \left( \frac{\Delta_{p,q} d^{2/q}}{\gamma^2} + \ln \frac{4}{\alpha} \right) \cdot \frac{\sigma'_f}{p-1} \cdot N^{\frac{1-2p}{p}} - \frac{\sigma'_f}{p-1} \cdot N^{\frac{1-p}{p}} \tag{32}$$

$$\geq F(\mathbf{x}^*) \cdot \left( 1 - \frac{7}{\ln 2} \cdot \left( \frac{\Delta_{p,q} d^{2/q}}{\gamma^2} + \ln \frac{4}{\alpha} \right) \cdot N^{-\frac{1}{2}} \right)$$

$$- \frac{7}{\ln 2} \cdot \left( \frac{\Delta_{p,q} d^{2/q}}{\gamma^2} + \ln \frac{4}{\alpha} \right) \cdot N^{\frac{2-p}{2p}} - \frac{7}{\ln 2} \cdot \left( \frac{\Delta_{p,q} d^{2/q}}{\gamma^2} + \ln \frac{4}{\alpha} \right) \cdot \frac{\sigma'_f}{p-1} \cdot N^{\frac{2-p}{2p}} - \frac{\sigma'_f}{p-1} \cdot N^{\frac{2-p}{2p}} \tag{33}$$

$$\geq F(\mathbf{x}^*) \cdot \left( 1 - \frac{7}{\ln 2} \cdot \left( \frac{\Delta_{p,q} d^{2/q}}{\gamma^2} + \ln \frac{4}{\alpha} \right) \cdot N^{-\frac{1}{2}} \right) - \frac{8}{\ln 2} \cdot \left( \frac{\Delta_{p,q} d^{2/q}}{\gamma^2} + \ln \frac{4}{\alpha} \right) \cdot \left( 1 + \frac{\sigma'_f}{p-1} \right) \cdot N^{\frac{2-p}{2p}}, \tag{34}$$

where the first inequality (31) is due to (23), (27) and (28). Equation (32) is due to combining like terms. The inequality (33) is due to (29) and the fact that $\frac{1-2p}{p} \geq \frac{2-3p}{2p} \geq \frac{1-p}{p} \geq \frac{2-p}{2p}$, and further combining like terms leads to (34). $\square$

## D. Auxiliary Lemma

**Lemma D.1.** *Let $p \in [2, \infty)$. Denote by $\xi_1, ..., \xi_N \in \mathbb{R}$ an i.i.d. sequence of random variables with $\mathbb{E}[\xi_1] = 0$. Then $\left\| N^{-1} \sum_{j=1}^N \xi_j \right\|_{L^p} \leq 2.74 \cdot 2^{1/p} \cdot \sqrt{pN^{-1}} \cdot \|\xi_1\|_{L^p}$.*

*Proof.* This lemma is largely based on the proof embedded in Proposition 1 by Oliveira & Thompson (2023). By Marcinkiewicz' inequality, for some universal constant $K < 0.935$,

$$\left\| N^{-1} \sum_{j=1}^N \xi_j \right\|_{L^p}$$

$$\leq N^{-1} 2^{1+1/p} \sqrt{2K} \cdot \sqrt{p} \left\| \sum_{j=1}^N \xi_j^2 \right\|_{L^{p/2}}^{1/2} \leq N^{-1} 2^{1+1/p} \sqrt{2K} \cdot \sqrt{p} \sqrt{\sum_{j=1}^N \left\| \xi_j^2 \right\|_{L^{p/2}}}$$

$$= N^{-1} 2^{1+1/p} \sqrt{2K} \cdot \sqrt{p} \sqrt{\sum_{j=1}^N \|\xi_j\|_{L^p}^2} = 2^{1+1/p} \sqrt{2K} \cdot \sqrt{p \cdot N^{-1}} \cdot \|\xi_1\|_{L^p}.$$

as desired. $\square$

# E. Additional Detail on Numerical Experiments

## E.1. Estimation of Problem Quantities

Problem quantities in P-NCB$^E$ were over-estimated to mimic more realistic applications with limited knowledge of the problem. In the experiment results presented in Table 1 and Table 3, for the stochastic linear problem subject to a simplex, one may see that over-estimation of the aforementioned quantities like $\sigma_f$ and $\sigma_g$ are accessible. More specifically, if we denote $\boldsymbol{\kappa} := (\kappa_1, \ldots, \kappa_d)$, then for $\mathbf{x} \in \mathcal{X}$, it holds that $\sigma_f = \|f(\mathbf{x}, \xi) - F(\mathbf{x})\|_{L^p} = \left\|\sum_{i=1}^{d} \kappa_i(\xi_i - \mathbb{E}[\xi_i])x_i\right\|_{L^p} \leq$

$\|\boldsymbol{\kappa}\|_\infty \mathbb{E}\left[\|\xi_1 - \mathbb{E}[\xi_1]\|_\infty^p\right]^{1/p}$ and $\sigma_g = \|g_f^*(\xi) - g_F^*\|_{L^p} = \mathbb{E}\left[\left|\sum_{i=1}^{d} \kappa_i(\xi_i - \mathbb{E}[\xi_i])\right|^p\right]^{1/p}$. The expected values herein can be further estimated using Monte Carlo simulation performed on an independent validation set of (no more than) the same number of sample points in the SAA formulation.

For problem quantities in P-NCB-NC$^E$, we set $\max_{\mathbf{x} \in \mathcal{X}} \|\mathbf{x}\|_p^2$ to be 1, and we sample 100 sample points $\mathbf{x}_i, i = 1, \ldots, 100$ uniformly inside $\mathcal{X}$ to be further estimate $\sigma_f = \|f(\mathbf{x}, \xi) - F(\mathbf{x})\|_{L^p}$, using Monte Carlo simulation performed on an independent validation set of (no more than) the same number of sample points in the SAA formulation, and we choose the maximum from these 100 sampled estimates. For $\sigma_f'$ in the upper bound, we used the uniformed quantities $\sigma_f$ as an over-estimation of $\sigma_f'$.

## E.2. Solving SAA and DRM Problem

For the stochastic linear problem, the optimal solution of the SAA formulation admits a straightforward closed form. For the over-parameterized model, the SAA solution is achieved using stochastic gradient descent, where the learning rate is specified as $5 \times 10^{-7}$, and maximum iteration number 100. The DRM solution is achieved using Algorithm 1 in Norton & Royset (2023), where $|B_t|$ therein is specified as 200, $\gamma$ is specified as 0.01, learning rate is specified as $1 \times 10^{-6}$, and maximum iteration number is 1,500.

## E.3. Additional Numerical Results

Problem quantities in P-NCB$^E$ were over-estimated to mimic more realistic applications with limited knowledge of the problem, whereas in P-NCB$^*$ the true values of these quantities were used.

The ECP values in Table 3 shows the varying tightness among P-NCB$^*$, P-NCB$^E$ and B-NCB. B-NCB consistently achieves full coverage, indicating overly conservative bounds. P-NCB$^*$ and P-NCB$^E$ results attain slightly lower yet valid coverage, reflecting relatively less conservative bounds.

*Table 3.* ECP comparisons among P-NCB$^*$, P-NCB$^E$ and B-NCB across dimensions and sample sizes for convex SO

| Sample Size (N) | Method | dim(d)=100 | dim(d)=500 | dim(d)=1000 | dim(d)=2000 | dim(d)=4000 |
|---|---|---|---|---|---|---|
| | P-NCB$^*$ | 1.0000 | 1.0000 | 1.0000 | 1.0000 | 0.9998 |
| 5 | P-NCB$^E$ | 1.0000 | 1.0000 | 1.0000 | 1.0000 | 0.9998 |
| | B-NCB | 1.0000 | 1.0000 | 1.0000 | 1.0000 | 1.0000 |
| | P-NCB$^*$ | 1.0000 | 1.0000 | 1.0000 | 0.9998 | 0.9998 |
| 10 | P-NCB$^E$ | 1.0000 | 1.0000 | 1.0000 | 0.9998 | 1.0000 |
| | B-NCB | 1.0000 | 1.0000 | 1.0000 | 1.0000 | 1.0000 |
| | P-NCB$^*$ | 1.0000 | 0.9998 | 0.9998 | 0.9999 | 1.0000 |
| 50 | P-NCB$^E$ | 1.0000 | 0.9998 | 1.0000 | 1.0000 | 1.0000 |
| | B-NCB | 1.0000 | 1.0000 | 1.0000 | 1.0000 | 1.0000 |
| | P-NCB$^*$ | 1.0000 | 0.9998 | 0.9999 | 1.0000 | 1.0000 |
| 100 | P-NCB$^E$ | 1.0000 | 1.0000 | 1.0000 | 1.0000 | 1.0000 |
| | B-NCB | 1.0000 | 1.0000 | 1.0000 | 1.0000 | 1.0000 |
| | P-NCB$^*$ | 0.9998 | 1.0000 | 1.0000 | 1.0000 | 0.9999 |
| 500 | P-NCB$^E$ | 1.0000 | 1.0000 | 1.0000 | 1.0000 | 1.0000 |
| | B-NCB | 1.0000 | 1.0000 | 1.0000 | 1.0000 | 1.0000 |
| | P-NCB$^*$ | 0.9999 | 1.0000 | 1.0000 | 0.9999 | 1.0000 |
| 1000 | P-NCB$^E$ | 1.0000 | 1.0000 | 1.0000 | 1.0000 | 1.0000 |
| | B-NCB | 1.0000 | 1.0000 | 1.0000 | 1.0000 | 1.0000 |

Table 4 shows the comparison results between the proposed method (P-NCB-NC$^E$) and the benchmark method (B-NCB-NC$^E$) in terms of ECP. We observed a consistently high ECPs, which indicated the correctness of both methods. Nonetheless, because our confidence level was $1 - \alpha = 0.9$, which was noticeably lower than the ECPs. This indicated a remaining level of conservatism of both our proposed and benchmark NCPs. We leave the further refining of NCPs to future work.

*Table 4.* Comparisons between P-NCB-NC$^E$ and B-NCB-NC$^E$ in ECP under various combinations of $N$ and $d$ for nonconvex SO

| $N$ | $d$ | P-NCB-NC$^E$ | B-NCB-NC$^E$ |
|---|---|---|---|
| 300 | 41 | 1.0000 | 1.0000 |
| 340 | 41 | 1.0000 | 1.0000 |
| 380 | 41 | 1.0000 | 1.0000 |
| 420 | 41 | 1.0000 | 1.0000 |
| 500 | 961 | 1.0000 | 1.0000 |
| 600 | 961 | 1.0000 | 1.0000 |
| 700 | 961 | 1.0000 | 1.0000 |
| 800 | 961 | 1.0000 | 1.0000 |
| 500 | 1681 | 1.0000 | 1.0000 |
| 600 | 1681 | 1.0000 | 1.0000 |
| 700 | 1681 | 1.0000 | 1.0000 |
| 800 | 1681 | 1.0000 | 1.0000 |

