# OpenReview forum: "Non-Asymptotic and Non-Lipschitzian Bounds on Optimal Values in Stochastic Optimization Under Heavy Tails"
_ICML.cc/2025/Conference — ICML 2025 poster_

### Official Review · Reviewer_sS3B · 2025-02-20

**Overall Recommendation:** 4

**Summary:**

This paper aims at proving confidence bounds over the minimum of a stochastic optimisation problem, ie, find a high probability confidence interval for the value of $F(x^\star) = \min E[f(x,\xi)]$, given sampled datapoints $(\xi_1,\dots,\xi_N)$. Thus, this task typically boils down to proving a lower bound and an upper bound. While significant work has been done on this topic, existing work often either have a problematic dependence in the Lipschitz constant of $f$ or the metric entropy of the domain (which might explode in practice) or require light-tail assumptions and / or convexity.
The present work address these issues by proving three new non-asymptotic confidence bounds (NCB): (i) a NCB for non-Lipschitz convex problems under heavy-tailed assumptions, (ii) for the first time a non-Lipschitz and non-convex NCB with light-tail (sub-exponential) assumptions, and (iii) a NCB in a non-convex heavy-tailed setting, at the cost of additional assumptions.
The proof techniques rely on two existing techniques to construct confidence bounds, namely the sample average approximation (SAA) and the diametrical empirical risk minimisation (DRM), which are extended beyond their classical assumptions.

**Claims And Evidence:**

The three main theoretical claims (ie, the three new confidence bounds mentioned in the summary above) are clearly presented. Full proofs are provided in the appendix (see the theoretical claims section).

To improve clarity, it could have been helpful to quickly discuss in the main text the origin of the constants appearing in the bound, especially the ones coming from the Marcinkiewicz-Zygmund inequality, used in the appendix, and from the moments of the Gaussian distribution.

**Essential References Not Discussed:**

N/A

**Experimental Designs Or Analyses:**

N/A

**Methods And Evaluation Criteria:**

To construct their new non-asymptotic confidence bounds, the authors build on the existing SAA and DRM techniques, the approach is well-motivated and make sense in this context.
The main assumptions are made on the tail of the distribution of $f$ (and a convexity assumption for theorem 4.1) and its gradient and are pertinent with the problem.
Assumption 4.8, on which rely the last result, seems significantly stronger. Even though the authors quickly argue that it can be satisfied in machine learning settings, it would probably require more discussion and/or examples.

**Other Comments Or Suggestions:**

Were you aware that you were allowed 8 pages of main text?

Beside that, here a a few minor remarks:
 - $\sigma_g$ is mentioned in the introduction but the notation for the gradients have not yet been introduced.
 - Line 184 second column: the $\forall x \in \mathcal{X}$ does not mean anything, as there is already $\inf_{x \in \mathcal{X}}$.

**Other Strengths And Weaknesses:**

In general, I think that the limitations (finite diameter, assumption 4.8, ...) could be more largely discussed, especially since one extra page as allowed.

**Questions For Authors:**

1. You proofs strongly rely on the assumption that the domain $\mathcal{X}$ has bounded diameter, is it possible to relax this assumption? Are there negative results saying that this assumption is not necessary. In general, I think this assumption should be discussed a bit more as it is not clear that it can be satisfied in practical settings.

 2. Regarding remark 4.3, does this mean that your results imply the light-tailed results? Is it true under the same assumptions as the existing light-tailed result? Are the constants worse?

**Relation To Broader Scientific Literature:**

The paper is related to existing literature on confidence bounds, sample average approximation and diametrical empirical risk minimisation, which seems to be well-discussed by the authors, even though it is not my main domain of expertise.

**Theoretical Claims:**

I did review most of the theoretical claims apart from the details of the proof of Theorem 4.9. I did not notice any obvious mistake.

I have only one remark: in the introduction, when discussing the main results 2, you mention that the growth of $\Delta_{p,q}$ leads to a $\mathcal{O}(\log d)$ dependence in the bound, this is correct but I did not see a proof, it could be worth adding it to the discussion.

---

> ### Author Rebuttal · Authors · 2025-03-27
>
> We appreciate the reviewer's very careful and insightful evaluation.
>
> 1. Re: "Other Comments": We will make sure to make full use of the 8 page limit. To that end, we will present part of our additional numerical results and add more discussions in response to the comments of this reviewer. We will also fix the minor remarks in the full paper.
>
> 2. We performed numerical experiments on two sets of problems. (i) for convex problems, we considered a stochastic linear program; (ii) for nonconvex problems, we considered a neural network for regression. Please see more detail about our partial results on the convex case in response to reviewer [ptuU]. Here, we only present some partial results for the nonconvex case. More specifically, we considered the problem of training a two-layer neural network with a data generating process $Y=g(X)+\epsilon$, where $g$ is some unknown function in nature to be reconstructed through the observations of some $N$-many sample points of $(X,Y)$. For the purpose of simulation, we specified $g$ to be a two layer neural network with LeakyRulu activation function and randomly simulated weights. Here, $\epsilon$ is assumed to follow student $t$-distribution to simulate heavy-tailed underlying randomness. We tested the SAA and DRM problem corresponding to training a neural network in minimizing mean squared error with different combinations of the sample size $N$ and problem dimensionality $d$. We then compared the width of the region between confidence bounds (CB) under significance level $\alpha=0.1$ with benchmark (referred to as "Benchmark") by Oliveira and Thompson (2023). Table 3 below present the average ratio of the widths between our proposed CB and the benchmark CB, calculated as $(U_{proposed}-L_{proposed})/(U_{Benchmark}-L_{Benchmark})$, averaged out of 10 independently random replications. Here $U$ and $L$ denote the upper confidence bound and lower confidence bound, respective to their subscripts.
>
> **Table 3**: Average ratio of the length of proposed CB and benchmark CB. (A number smaller than 1 means that the proposed CB has a smaller width and thus is better).
> |Sample Size \(N\) | dim\(d\)=441 | Sample Size \(N\) | dim\(d\)=961 | Sample Size \(N\)| dim\(d\)=1681  |
> |------------|--------|---------|---------|---------|---------|
> | N = 300 | 0.0296 | N = 500 | 0.0230 | N=500 | 0.0093 |
> | N = 340 | 0.0182 | N = 600 | 0.0116 | N=600 | 0.0087 |
> | N = 380 | 0.0176 | N = 700 | 0.0119 | N=700 | 0.0089 |
> | N = 420 | 0.0166 | N= 800  | 0.0112 | N=800 | 0.0084 |
>
> 3. Re: "Methods And Evaluation Criteria": We will add more thorough discussions on this assumption. One illustration to include is that this assumption is directly related to the assumption of the presence of a positive margin in separating the data population in the application of classification. E.g., Assumption 3.3 in Yuan and Gu (2020). Generalization error bounds of gradient descent for learning over-parameterized deep relu networks. (AAAI). In this example, the assumed constant margin therein can imply the said perturbed settings.
>
> 4. Re: "Claims & Evidence": We will add discussions in the full paper to illustrate the origin of the constants. In particular, "2.74" originates from the Marcinkiewicz's inequality.
>
> 5. Re: "Theoretical": We had a typo in this mentioned statement. We meant to claim that the quantity grows at the rate of $O(\ln d)$, when the true solution satisfies the problem structure that the solution lies within a $1$-norm ball with dimension-independent radius. This is the common assumption of weak sparsity in statistics and applied optimization. Under this assumption, the dependence on $d^{2/q}$ can be bounded by a constant as $q$ can be specified as $\ln d$. In the same case $\Delta_{p,q}$ would only grow at the rate of $O(\ln d)$. We are sorry about this oversight and will carefully discuss this in the revision.
>
> 6. Re: Q1: Benchmark results (e.g., Guigues et al., 2017, among others) all assume a bounded feasible region. In most practical problems, there exists at least one finite optimal solution $\mathbf x^*$. When this happens, one may use the distance $D$ from $\mathbf x^*$ to a known feasible solution $\mathbf x_0$ as the diameter of the feasible region (since the problem remain equivalent if one were to impose an additional constraint that $\mathbf x:\,\Vert \mathbf x-\frac{\mathbf x_0+\mathbf x^*}{2}\Vert_q\leq \mathcal D$. The question then boils down to how to estimate $\mathcal D$. To that end, one often may resort to cross-validation. The bounded assumption is also practical in training neural networks as one could expect weights normalization.
>
> 7. Re: Q2: Our results imply the same rate as the light-tailed results. Yet, to accommodate for heavy tails, the constants are relatively elevated; there is an addition of constant $K\leq 2.74$, which is in the result of Marcinkiewicz’ inequality. This quantity can hardly be improved without fundamental improvement on Marcinkiewicz’.

---

> > ### Comment · Reviewer_sS3B · 2025-04-02
> >
> > Thank you very much for taking the time to answer my questions and performing additional experiments.
> > I will keep my score.

---

> > > ### Author Response · Authors · 2025-04-04
> > >
> > > We thank the reviewer for the careful evaluation, insightful comments, and encouraging remarks.

---

### Official Review · Reviewer_LeHx · 2025-03-03

**Overall Recommendation:** 3

**Summary:**

The paper studies confidence bounds for the optimal value of a stochastic optimization problem. For the convex case, the paper derives bounds for a set of heavy-tailed assumptions, and the bounds do not depend on the Lipschitz constant of the function. For the non-convex case, the paper provides bounds for both light tailed and heavy-tailed assumptions, from a solution obtained via a DRM formulation.

**Claims And Evidence:**

- The claims in the paper seem all sound.

**Essential References Not Discussed:**

N/A

**Experimental Designs Or Analyses:**

N/A.  The paper is theoretical and doesn't require empirical evaluation.

**Methods And Evaluation Criteria:**

N/A. The paper is theoretical and doesn't require empirical evaluation.

**Other Comments Or Suggestions:**

N/A

**Other Strengths And Weaknesses:**

It's also hard to evaluate the dependence of the bounds on some of the parameters such as the dimensions and the $\Delta_{p,q}$ term. Are there any lower bounds saying what the best CB we can achieve?

**Questions For Authors:**

My general question about this paper is that, instead of a blackbox solution of the problem on the samples like in Algorithm 1, can there be any improvements to the bounds if we know also the stochastic optimization algorithm, such as SGD? SGD doesn't really fit in the framework of Algorithm, but we also have bounds in high probability, even in the heavy tailed case.

**Relation To Broader Scientific Literature:**

I find it hard to appreciate the results in the paper. More precisely, I don't really understand the challenge to derive bounds in this paper. Once having assumptions 3.1 and 3.2, and the assumption that we have the optimal solution on the samples, it's conceivable that using concentration inequalities related to $p$-moments (if we care about heavy tails) should be sufficient (here it's Marcinkiewicz's inequality in appendix D).

**Theoretical Claims:**

I checked some of the proofs in the appendix (Thm 4.1 and 4.5) and didn't see any problems, although some of the details should be mentioned more clearly. For example, the $\epsilon$-net argument in Theorem 4.5 should be explained in more detail.

---

> ### Author Rebuttal · Authors · 2025-03-27
>
> We thank the reviewer for in-depth evaluation and insightful comments. Below citations follow the same reference list as in our paper.
>
> 1. To answer the comment re: "[...]I don't see the challenge to derive bounds[...]", we would like to point out that we made conscious and non-trivial effort to simplify the proof so as to make it better verifiable and readable. Bounding optimal values has been a research question pursued by much literature, and heavy-tails are traditionally known challenge to related problems. For instance, earlier work on asymptotic results dates back to, e.g., King & Rockafellar (1993), among many others, and more recent work on non-asymptotic results include those presented Guigues et al. (2017) and Oliveira & Thompson (2023). We would hope to use these references as evidence to showcase that our research question (even in the case of convex problems) is not a trivial one. Indeed, to our knowledge, whether non-Lipschitzian and heavy-tailed bounds for optimal values are admissible even in the convex case has not been well understood thus far. After we connect the dots between the research question and some known mathematical tools (Marcinkiewicz's inequality), it was also to our surprise that our proposed answer and its proof would appear to be straightforward.
>
> 2. We are not aware of a reference that explicitly shows lower bounds on the major quantities in our setting (e.g., non-asymptotic, non-Lipschitz, and heavy-tailed). In the convex case, our rate is almost exactly the same as the benchmark bounds provided by Guigues et al. (2017) --- the only difference is that the dependence on the significance level $\alpha$ is $\ln(1/\alpha)$ in the results of Guigues et al. (2017), while, in contrast, our results grow at the rate of $O(\alpha^{-2/p})$. This change would be unavoidable since Guigues et al. (2017) deal with light-tailed problems whose tail is characterized with an exponential decay. Meanwhile, our focus is on heavy-tailed settings where central moments exist up to the $p$-th order. In nonconvex case, our results do not depend on Lipschitz constant and thus can often provide sharper (narrower) confidence intervals. (In this regard, please see Tables 1, 2 and 3 for some numerical comparisons provided in our responses to Reviewer 1/ptuU and Reviewer 4/sS3B).
>
> 3. $\Delta_{p,q}$ shows up only in our results for nonconvex case. Explications of this quantity is provided in Remark 4.6. As long as we can (coarsely) estimate (i) some estimate on the order of some existent central moment of the underlying randomness, and (ii) the diameter of the feasible region, then $\Delta_{p,q}$ can be estimated. The quantities to be estimated are no more than the benchmark results such as Guigues et al. (2017) and Oliveira & Thompson (2023).
>
> 4. While many results on SGD has focused on asymptotics, SGD does provide non-asymptotic bounds for optimal values, as is studied by Lan et al. (2012). To be consistent with Lan et al. (2012), we reviewed those results under the (alternative) name of mirror descent stochastic approximation (MDSA) in Section 2 of our manuscript. Our SAA-based bounds improves over the MDSA in terms of the dependence on Lipschitz constant.  Furthermore, confidence bounds provided by Lan et al. (2012) apply to either the case on the two ends of a distribution spectrum --- namely, (i) the case where only the second moment of the underlying randomness is bounded; and (ii) the case where the underlying randomness is light-tailed (sub-gaussian). Our findings additionally provide results for cases where the underlying randomness has existent central moments up to the $p$th order for $p\geq 2$. One may also observe that our bounds present identical rates as MDSA in terms of the dependence on some critical quantities (other than Lipschitz constants), such as sample size and dimensionality. So, perhaps a shorter answer would be that SGD is not known to improve the non-asymptotic confidence bounds from the literature; instead, the existing results seem to only show that SGD-based non-asymptotic confidence bounds would be comparably less appealing.

---

### Official Review · Reviewer_nMkb · 2025-03-04

**Overall Recommendation:** 4

**Summary:**

The paper presents non-asymptotic bounds on the minimal value of a stochastic optimization problem. The novel contributions are that the resulting bounds do not depend on global Lipschitz constants of the integrand function or the objective and operate for heavy-tailed function value and gradient distributions.

**Claims And Evidence:**

The paper's claims are theoretical and they are proven soundly.

**Essential References Not Discussed:**

I believe at least some of the paper should be related to confidence bounds that arise from min-max methods similar to DRM. A seed reference (and references therein) I would like to see discussed is Duchi et al (2021). While I realize that DRM perturbs the parameters and not the inputs or input distributions, it would be worth it to discuss the technical differences (proof techniques, etc) of these other approaches to confidence bounds.

John C. Duchi , Peter W. Glynn , Hongseok Namkoong (2021) Statistics of Robust Optimization: A Generalized Empirical Likelihood Approach. Mathematics of Operations Research.

**Experimental Designs Or Analyses:**

There are no experiments in the paper.

**Methods And Evaluation Criteria:**

They do make "sense", although, the main issue I find is that there is no discussion of estimating the problem constants that appear in the confidence bounds. Namely, this includes $\sigma_f$, $\sigma_g$, $\mathcal{D}\_q$, $\sigma\_\psi$, etc.

**Other Comments Or Suggestions:**

This is minor, but the computations on page 12 and 13 can be made a little easier to follow, perhaps with colors or by breaking up the long blocks of display equations.

**Other Strengths And Weaknesses:**

**Strengths:** The paper is remarkably well-written and logically ordered.

**Weaknesses:**
My main weaknesses are summarized above.
- A confidence bound paper with no experiments and no ability to estimate the relevant constants seems below the bar, and this is my main concern. While there is clearly a theoretical focus, experiments are not a superfluous detail in this setting, which is motivated by assessing algorithm performance. I would increase the score significantly if these two points were addressed.
- If metric entropy is discussed, then I believe that the dimension dependence (i.e.~on $d$) should be shown in the main results.

**Questions For Authors:**

Can you explain why the bounds in Theorem 4.1 (and other results) grow with a factor $\sqrt{p}$? Why do the intervals widen to infinity of the random variables are nearly bounded? I see this stems from D.1, but additional explanation would be helpful.

**Relation To Broader Scientific Literature:**

Yes, in that the bounds avoid metric entropy and global Lipschitz factors. However, I find these claims slightly overstated. I believe the non-Lipschitzian aspect can be framed as having bounds that depend on the average gradient norms (or their deviations) as in Assumption 3.1, as opposed to the maximum gradient norms, which more clearly highlights the technical differences. Furthermore, the part that I find more strange is the discussion of metric entropy as an undesirable dependence, but the results of Theorem 4.5 and 4.9 include explicit dimension factors which are hidden in the statement of the results. It is not clear that these factors scale any better (or possibly worse) with metric entropies of various feasible sets.

**Theoretical Claims:**

I read through the proofs briefly, and paid closer attention to Appendix C due to the interest of the bounded surrogate argument for extending light-tailed analyses to possibly heavy-tailed distributions.

---

> ### Author Rebuttal · Authors · 2025-03-27
>
> We thank the reviewer for insightful comments.
>
> 1. Re: Relation To Literature: Average/expected norm of gradient is essentially the (non-central) moment, which typically has two components: **(A)**. the norm of the gradient of the expected cost function, and **(B)**. the central moment of the gradient of the random cost function. The Lipschitz constant of a (population-level formulation of) stochastic program is associated with **(A)**, instead **(B)**. Assumption 3.1 imposes upper bound on component **(B)**. One can easily construct cases where the Lipschitz constant grows while **(B)** remain unchanged.
>
> 2. Theorem 4.1 is free from metric entropy; yet, we do NOT claim Theorems 4.5 and 4.9 are free from metric entropy. We will emphasize this more in revision. We explicate (and do not hide) dependence on $d$ in all our mentioning of Theorems 4.5 and 4.9. Our results do not depend on Lipschitz constants, which is often a critical quantity that may grow (rapidly) with $d$.
>
> 3. We will add detailed discussions on Duchi et al. (2021) and related references. As mentioned in Section 2, existing confidence bounds are either asymptotic or non-asymptotic. The focus of our paper is on the latter. In contrast, Duchi et al. (2021) provide novel asymptotic results based on DRO. We will also carefully discuss the differentiation as suggested by the reviewer.
>
> 4. Re Weakness on experiments:  We performed experiments on two sets of problems. (i) for convex problems, we considered a stochastic linear program; (ii) for nonconvex problems, we considered a neural network for regression, which is known to entail data that represent potential heavy-tails in the underlying randomness. Please check our response to reviewer [ptuU] and reviewer [sS3B] for detailed experiment settings. The full paper will also show representations of the results in plots, where one can observe a strong scalability as dimensionality increases.
>
> 5. Re Weakness on estimating quantities: We would like to argue that the benchmark papers have not provided generic schemes to estimate the quantities either. For instance, in the numerical examples by the benchmark (Guigues et al., 2017), estimation of the tail parameters is made possible in an ad-hoc fashion by exploiting the assumption that the underlying distribution is uniform (and thus of bounded support set). Nonetheless, we agree that proper estimation of these quantities will make the results significantly better useful. One common (expedient) approach is to use sample average values of quantities evaluated at a computed solution to the SAA formulation. E.g., we may evaluate $\sigma_g$ (and $\sigma_f$) by calculating the sample $p$-norm distance of gradient (and objective function value, resp.) from their sample mean at a (near-)optimal solution to SAA. This approach has been employed in estimating quantities in linear regression (often, those quantities are special cases of $\sigma_g$ and/or $\sigma_f$).
>
> 6. Related to 5, another approach is to use the knowledge of the problem structure to (over)estimate the quantities. Indeed, in the experiment presented in response to Review [ptuU] for a stochastic linear program subject to a simplex, one may see that over-estimation of the aforementioned quantities is accessible. More specifically, for $\mathbf{x}\in\mathcal {X}$, $\Vert f(\mathbf x,\xi)-F(\mathbf x)\Vert_{L^{p}}=\Vert\sum_{i=1}^d\kappa_i (\xi_i-\mathbb E[\xi_i])x_i\Vert_{L^{p}}\leq\Vert\kappa\Vert_{\infty}\Vert (\xi-\mathbb E[\xi])^T\mathbf{x}\Vert_{L^{p}}\leq\Vert\kappa\Vert_{\infty}\Vert \Vert(\xi-\mathbb E[\xi])\Vert_{\infty}\Vert\mathbf{x}\Vert_1\Vert_{L^{p}} = \Vert\kappa\Vert_{\infty}\Vert\Vert(\xi-\mathbb E[\xi])\Vert_{\infty}\Vert_{L^p}$ and $\Vert g_f^*(\xi)-g_F^*\Vert_{L^{p}}=\mathbb E[\sum_{i=1}^d\vert\kappa_i(\xi_i-\mathbb E[\xi_i])\vert^p]^{1/p}=\Vert\kappa\Vert_p\mathbb E[(\xi_1-\mathbb E[\xi_1])\vert^p]^{1/p}$. The expected values herein can be estimated using Monte Carlo performed on an independent validation set of (no more than) the same number of sample points in the SAA formulation. $p$ can be determined based on its best performing choice; since our confidence bounds are effective for all admissible $p$, one may use the value of $p$ that yields the smallest width of the output confidence interval(s).
>
> 7. Our convex results are metric entropy-free. Comparable pattern is presented by Guigues et al. (2017). Our nonconvex result explicates the dependence on dimensionality $d$ as $d^{-2/q}$.
>
> 8. Re Questions for authors: The presence of $\sqrt{p}$ is often due to the use of Marcinkiewicz's inequality. Our results hold for all admissible choices of $p\geq2$; namely, one may choose $p$ from all possible values the one that would lead to the smallest width of the confidence interval.  As in Remark 4.3, under light-tailed-ness (and thus all central moments exist) $p$ could be specified as $\ln(6/\alpha)$ to recover the same rate as per Guigues et al. (2017) in light-tailed setting.

---

### Official Review · Reviewer_ptuU · 2025-03-14

**Overall Recommendation:** 4

**Summary:**

The paper provides non-asymptotic confidence intervals for the solutions of stochastic optimization problems. Unlike previous work, their approach simultaneously covers non-Lipschitz and heavy-tailed problems. They also include analysis of non-convex and overparametrized cases.

**Claims And Evidence:**

Claims are well supported.

**Essential References Not Discussed:**

I am not very familiar with the literature

**Experimental Designs Or Analyses:**

N/A

**Methods And Evaluation Criteria:**

N/A

**Other Comments Or Suggestions:**

The summarized title in the header of each page remains unchanged from the template, please correct that.

**Other Strengths And Weaknesses:**

The paper is clearly written and related work seems properly acknowledged. I believe the contributions are relevant. However, I miss evaluations of the confidence intervals in (at least) some toy example in order to verify their tightness. Are there any difficulties for evaluating the bounds in specific SO problems?

**Questions For Authors:**

See *Strengths And Weaknesses*

**Relation To Broader Scientific Literature:**

The paper relaxes the assumptions for the analysis of stochastic optimization problem, which has potential impact in several subfields.

**Theoretical Claims:**

The proofs in the Appendix seem correct.

---

> ### Author Rebuttal · Authors · 2025-03-27
>
> We appreciate the great effort by the review in evaluating our manuscript. All papers cited in this response are the same as those included in the original submission.
>
> 1. Following the comments, we conducted two experiments on a convex and a nonconvex problem. For the former, we considered a stochastic linear program; and, for the latter, we considered a neural network in regression. In the revision of the paper, we will formally present these tables and plot out critical trends. In strategizing spacing, we only present our partial results for convex case here; results on nonconvex case are in our response to reviewer [sS3B]. We considered a stochastic minimization problem with constraints ${\mathbf x = (x_1,\cdots,x_d) : \sum_{i=1}^d x_i =1, \mathbf x\geq0}$ with a stochastic cost function $f(\mathbf x,\xi):= -\sum_{i=1}^d\kappa_i\xi_i x_i$, where we let $\kappa_i =0.08 +\frac{0.04(i-1)}{d}$ for $i = 1,2,\cdots,d$, and each $\xi_i$ is a power law distributed random variable with density function $p_{\xi}(x)=\frac{ab^a}{x^{a+1}}$ for $x>b$. Here $a>0$ and $b>0$ are two parameters in power law distribution; thus the highest order of existence of moments is $a-1$. In our experiments, we set $a=3.01, b=1$ (and correspondingly $p=2$) and tested for $10,000$ replications under the confidence level of $\alpha=0.1$. We presented the empirical coverage probability in Table 1, where the results generated by our approach is referred to as 'Proposed'. The empirical coverage probability is calculated as the proportion of replications in which $F^*$ lies within the confidence bounds, i.e. $L_{N,\alpha}\leq F^*\leq U_{N,\alpha}$.
>
> - We also tested the benchmark scheme (referred to as "benchmark" hereafter) by Oliveira & Thompson (2023) in the same manner as the above.
>
> **Table 1:**  Estimated Coverage Probability for $\alpha = 0.01, a = 3.01$ and $p=2$. A number closer to 0.99 is better. Note here, problem quantities in our bound were over-estimated to mimic more realistic applications with limited knowledge of the problem.
>
> | Sample Size \(N\) | Method | dim\(d\)=100 | dim\(d\)=500 | dim\(d\)=1000 | dim\(d\)=2000 | dim\(d\)=4000 |
> |-------------------|--------|---------|---------|---------|---------|---------|
> | N = 5 | Proposed | 1.0000  | 0.9999  | 0.9998  | 0.9999  | 0.9998  |
> | N = 5 | Benchmark | 1.0000  | 1.0000  | 1.0000  | 1.0000  | 1.0000  |
> |=============|========|=========|=========|=========|=========|=========|
> | N = 10 | Proposed | 1.0000  | 0.9999  | 0.9999  | 0.9998  | 0.9997  |
> | N = 10 | Benchmark | 1.0000  | 1.0000  | 1.0000  | 1.0000  | 1.0000  |
> |=============|========|=========|=========|=========|=========|=========|
> | N = 50 | Proposed | 0.9999  | 0.9998  | 0.9997  | 0.9999  | 1.0000  |
> | N = 50 | Benchmark | 1.0000  | 1.0000  | 1.0000  | 1.0000  | 1.0000  |
> |=============|========|=========|=========|=========|=========|=========|
> | N = 100 | Propose | 0.9999  | 0.9997  | 0.9997  | 1.0000  | 0.9999 |
> | N = 100 | Benchmark | 1.0000  | 1.0000  | 1.0000  | 1.0000 | 1.0000|
> |=============|========|=========|=========|=========|=========|=========|
> | N = 500 | Proposed | 0.9997  | 0.9999  | 1.0000  | 1.0000  | 0.9999 |
> | N = 500 | Benchmark | 1.0000  | 1.0000  | 1.0000  | 1.0000 | 1.0000 |
> |=============|========|=========|=========|=========|=========|=========|
> | N = 1000 | Proposed | 0.9998  | 1.0000  | 1.0000 | 0.9999 | 0.9998 |
> | N = 1000 | Benchmark | 1.0000  | 1.0000  | 1.0000  | 1.0000 | 1.0000 |
>
> - We also tested the length of confidence bound (CB) by averaging 10,000 replications, and we present the ratio of the length of our proposed CB and Benchmark CB. Please refer to our response to reviewer[sS3B] for the formal definition of length ratio.
>
> **Table 2:**  Average ratio between the width of proposed CB and Benchmark CB. (A number smaller than 1 means that the proposed CB has smaller width and thus is better).
> | Sample Size \(N\) | dim\(d\)=100 | dim\(d\)=500 | dim\(d\)=1000 | dim\(d\)=2000 | dim\(d\)=4000 |
> |-------------------|--------|---------|---------|---------|---------|
> | N = 5 |  0.3938  | 0.2390  | 0.2117  | 0.0991  | 0.0558  |
> | N = 10 |  0.3938  | 0.2390  | 0.2117  | 0.0991  |  0.0558  |
> | N = 50 |  0.3938 | 0.2390  | 0.2117 | 0.0991 | 0.0558  |
> | N = 100 | 0.3938  | 0.2390  | 0.2117  |0.0991 | 0.0558|
> | N = 500 | 0.3938  | 0.2390  | 0.2117 | 0.0991| 0.0558 |
> | N = 1000 | 0.3938 | 0.2390  | 0.2117 | 0.0991 | 0.0558 |
>
>
> As we can see both our proposed CB and Benchmark CB are both accurate. The Benchmark provides very loose CB with all tested coverage probability to be $1.000$, and the its widths are usually 4 times longer than our proposed CB. Our proposed CB, under the same heavy-tailed assumptions, can provide a tighter CB compared with Benchmark.
>
> 2. Response to "Other Strengths And Weaknesses": Please find details in estimation under our reply to reviewer [nMkb]-item 5&6.
>
> 3. Response to "Other Comments Or Suggestions": We will be sure to modify this in the full paper.

---

> > ### Comment · Reviewer_ptuU · 2025-04-04
> >
> > Thank you for the experiments, the tightness of the confidence bounds wrt to the benchmark of Oliveira & Thompson (2023) is a nice touch. I'll increase my score to 4.

---

> > > ### Author Response · Authors · 2025-04-04
> > >
> > > We thank the reviewer for carefully reviewing this paper and for the encouraging evaluation.

---

### Decision · Program_Chairs · 2025-05-01

**Decision:**

Accept (poster)

**Comment:**

This paper considers stochastic optimization and derives non-asymptotic confidence bounds for its solution. Authors study regimes that are not considered before, e.g. non-Lipschitzian confidence bounds for convex problems under heavy tails.


This paper was reviewed by four expert reviewers the following Scores: Accept, Accept, Accept, Weak Accept. I think the paper is studying an interesting topic and the results are relevant to ICML community. The following concerns were brought up by the reviewers:

- Dimension dependece of bounds

- No convincing experiments


Authors should carefully go over reviewers' suggestions and address any remaining concerns in their final revision. Based on the reviewers' suggestion, as well as my own assessment of the paper, I recommend including this paper to the ICML 2025 program.